# LuxS/AI-2 regulates *phoP/phoQ* by a non-canonical mechanism to enhance acid stress survival in *Salmonella* Typhimurium

Anmol Singh[1], Abhilash Vijay Nair[1☉¤], Shashanka Aroli[1☉], Suman Das[2], Subhrajit Karmakar[2], Raju S. Rajmani[3], Santanu Mukherjee[3], Umesh Varshney[1], Dipshikha Chakravortty◉[1,4]*

1 Department of Microbiology and Cell Biology, Division of Biological Sciences, Indian Institute of Science, Bengaluru, India, 2 Department of Organic Chemistry, Indian Institute of Science, Bengaluru, India, 3 Molecular Biophysics Unit, Indian Institute of Science, Bengaluru, India, 4 Adjunct Faculty, School of Biology, Indian Institute of Science Education and Research, Thiruvananthapuram, India

☉ These authors contributed equally to this work.
¤Current address: Department of Microbiology and Immunology, Vagelos College of Physicians and Surgeons, Columbia University, New York, New York, United States of America
* dipa@iisc.ac.in

## Abstract

The intestinal milieu is largely characterized by the complex array of chemical compounds produced through the metabolic activity of resident microbiota. Enteric pathogens like *Salmonella*, which have evolved refined mechanisms to persist within this environment, utilize these microbial metabolites and self-produce quorum molecules as molecular cues to identify ecological niches and modulate their survival and virulence strategies. *Salmonella* quorum sensing involves producing and detecting universal Autoinducer-2 (AI-2) signaling molecules. Our research reveals that *Salmonella* Typhimurium enhances AI-2 biosynthesis and transport under acidic conditions, aiding environmental adaptation and facilitating pathogenesis in macrophages. AI-2 signaling regulates the pH-sensing two-component system genes, *phoP/phoQ*, ensuring cytosolic pH homeostasis, survival, and acid tolerance. It also involves regulating the lysine/cadaverine-mediated acid tolerance response and maintaining bacterial cytosolic pH. Furthermore, we investigated the mechanism of AI-2-mediated gene regulation and demonstrated that, in addition to the *lsr* promoter, the repressor LsrR binds the *phoP* promoter via its Y25 and R43 residues, thereby negatively regulating *phoP* expression. Additionally, this signaling ameliorates the intracellular survival by modulating *Salmonella* Pathogenicity Island-2 (SPI-2) regulators (*ssrA/ssrB*) and SPI-2 effector expression via PhoP. Mouse models demonstrate that AI-2 signaling is essential for colonizing the primary and secondary infection sites. Therefore, quorum sensing facilitates survival in hostile host environments by modulating multiple genetic targets through the AI-2/LsrR-mediated feedback loop in pathogenic bacteria.

**Data availability statement:** All relevant data are in the manuscript and its supporting information files.

**Funding:** This work was funded by the Department of Biotechnology (DBT), Ministry of Science and Technology, Department of Science and Technology (DST), Ministry of Science and Technology to DC. DC acknowledges the Department of Atomic Energy -Science Research Council (DAE-SRC)(DAE00195) outstanding investigator award, ASTRA Chair Professorship, and TATA Innovation fellowship funds. The authors jointly acknowledge the DBT-IISc partnership program. Infrastructure support from ICMR (Center for Advanced Study in Molecular Medicine), DST (FIST), and UGC-CAS (special assistance) is acknowledged. AS acknowledges the UGC fellowship. AVN acknowledges the IISc-MHRD fellowship. SA acknowledges the CSIR fellowship. SD, SK, RSR, SM, and UV acknowledge financial support from IISc. The funders had no role in study design, data collection and analysis, decision to publish, or preparation of the manuscript.

**Competing interests:** The authors have declared that no competing interests exist.

## Author summary

To survive and cause disease within the host, enteric pathogens such as *Salmonella* sense and respond to changing environmental cues. One key strategy is quorum sensing, a cell–cell communication system. *Salmonella* uses the conserved quorum-sensing signal autoinducer-2 (AI-2) to coordinate adaptation and virulence. In this study, we show that *Salmonella* Typhimurium increases both AI-2 production and uptake under acidic conditions. AI-2 signaling promotes adaptation to hostile host environments by maintaining cytosolic pH homeostasis and enhancing survival inside macrophages. We demonstrate that AI-2 regulates major stress-response pathways, including the PhoP/PhoQ two-component system, and controls acid tolerance mechanisms involving lysine and cadaverine metabolism. We further identify a direct mechanism of AI-2–mediated gene regulation. The AI-2 responsive repressor LsrR binds not only to its canonical target but also to the *phoP* promoter, thereby fine-tuning stress responses. Through this regulatory network, AI-2 signaling modulates the expression of *Salmonella* Pathogenicity Island-2 genes required for intracellular survival. Using mouse infection models, we show that AI-2 signaling is essential for efficient colonization of both primary and secondary infection sites. Together, these findings establish quorum sensing as a central mechanism enabling *Salmonella* to survive hostile host environments and establish infection.

## Introduction

The community behaviour of bacteria, known as quorum sensing, protects them from various stress factors such as exposure to ultraviolet light, acids, detergents, or antimicrobial agents [1,2]. Gram-negative bacteria often regulate gene expression through the accumulation of autoinducers, such as N-acyl-homoserine lactones (AHLs or AI-1), mainly produced by Proteobacteria, some Bacteroidetes, Cyanobacteria, and Archaea. Gram-positive bacteria typically rely on autoinducing peptides (AIPs). In addition, autoinducer-2 (AI-2), a furanosyl borate diester, functions as a widely conserved interspecies signal [3]. Also, AI-2 signaling is linked to bacterial survival and pathogenesis, for example, in *Lactobacillus sp.,* AI-2 helps in survival under acidic conditions [4,5], and in *E. coli*, it plays a role in antibiotic resistance, pathogenicity, and even elicits the inflammatory pathway [6,7]. The effects of AI-2 are species-dependent, promoting biofilm formation in *Clostridioides difficile* but suppressing it in *Vibrio cholerae* [8]. In *Bifidobacterium breve*, *luxS* is essential for murine colonization and enhances iron acquisition, potentially increasing the pathogenicity of opportunists [9]. Furthermore, hosts can sense AI-2, trigger immune and inflammatory responses, and produce AI-2 mimics that activate bacterial QS, though the functional significance remains unclear [10].

At present, the QS is known to control and regulate an array of genetic targets in bacteria, thereby aiding and orchestrating bacterial life processes. For example, in *P.*

*aeruginosa,* and *Enterococcus faecalis* AI-2 signaling regulates the expression of virulence factors [11,12]. Also, in Enterohaemorrhagic *Escherichia coli* (EHEC) O157:H7, it activates the transcription of the two-component system, QseBC, and motility genes (*fliA* and *motA*) [13]. The LuxS/AI-2 signaling regulates the expression of oxidative stress response-related genes such as *sodA*, *sodCI*, and *sodCII*, which play an essential role in the survival of *Salmonella* within macrophages [14]. Further, the release of LsrR from DNA by AI-2 is essential for SPI-1 transcription and flagella expression in *Salmonella* [15,16]. Jesudhasan et al., 2010 determined that there is a differential gene expression between the STM WT and STM Δ*luxS* [17] pertaining to virulence, flagellar, and motility genes. Nonetheless, the molecular basis of QS regulation of genetic targets remains unclear, suggesting the existence of additional regulatory mechanisms.

*Salmonella* is considered a high-priority enteric pathogen worldwide [18], which has to thrive in diverse and competitive environments in the gastrointestinal tract. It needs to sense and adapt to a variety of environmental barriers, such as host physiology, nutrient availability, osmotic stress, and competition with other gut bacteria [19]. *Salmonella* synthesizes and recognizes the universal quorum-sensing signal AI-2 [20]. LuxS synthesises 4,5-dihydroxy-2,3-pentanedione (DPD), the universal precursor to AI-2 in *Salmonella*, which is maximally produced during its exponential growth phase [20,21]. The DPD molecule secreted by *Salmonella* and sensed by the LsrB protein is transported into the bacterial cytosol by an ABC transporter (LsrA, LsrB, LsrC, and LsrD) encoded by the *lsr* operon, and is phosphorylated by the LsrK kinase. The phosphorylated AI-2 binds the LsrR repressor protein and relieves the LsrR mediated repression of the *lsr* operon to enable further import of AI-2 (Fig 1A) [22,23]. Counteracting and conquering the acidic pH of the stomach and acidic intravacuolar environment is an indispensable survival strategy for *Salmonella.* We contemplated that the interbacterial communication and signaling systems might control the pH sensing and tolerance responses to aid in bacterial resilience. However, molecular basis of this regulation remains unclear, largely due to the limited understanding of AI-2 governed circuitry beyond the *lsr* operon.

Here, we show that LuxS/AI-2 signaling regulates a two-component system, *phoP/phoQ*, to activate the acid tolerance system in *Salmonella* and tune the SPI-2 encoded genes, enhancing its survival in *in vitro* and *in vivo* models. We also found that LsrR, the regulator of LuxS/AI-2 signaling, binds to the single and double strand of the *phoP/phoQ* promoter DNA via its Y25 and R43 amino acid residues, apart from its known regulation of the *lsr* operon. Thus, LuxS/AI-2 coordination facilitates dynamic changes of the transcriptome according to the surrounding environment and promotes rapid adaptation to different environmental niches to increase population fitness via an unrecognized circuitry in pathogenic bacteria.

## Results

### LuxS/AI-2 quorum sensing is required by *Salmonella* Typhimurium to survive in the macrophages

To begin our study, we first investigated the levels of AI-2 produced at different growth phases from *Salmonella* Typhimurium (STM) WT grown at 37°C in LB medium by using the *Vibrio harveyi* BB-170 reporter strain (which produces luminescence responses to AI-2). Consistent with previous studies [16], AI-2 dependent light production by *V. harveyi* BB-170 is maximum in the spent medium of the mid-log phase culture of *Salmonella* Typhimurium, and it gradually decreases towards the late stationary phase (S1A and S1B Fig). Next, we determined the mRNA expression level *in vitro* growth in LB medium, and we observed that while *Salmonella* upregulated the expression of the genes *lsrB*, *lsrK*, and *lsrR* during its mid-log phase to early stationary phase of growth (S1C Fig), it did not alter the expression of *luxS* during this period in LB medium (S1D Fig). And the deletion of *luxS* and genes (*lsrK*, *lsrB*, *lsrR*) in *Salmonella* did not affect the *in vitro* growth in LB and minimal medium (S1E and S1F Fig). Next, we examined whether AI-2 signaling influences the infection of *Salmonella* in macrophages. We first assessed bacterial uptake by RAW 264.7 macrophages and observed that, compared to STM WT, the quorum-sensing mutants STM Δ*luxS*, STM Δ*lsrB*, and STM Δ*lsrK* exhibited increased phagocytosis (Fig 1B). In contrast, the STM Δ*lsrR* mutant and the *luxS*-complemented STM Δ*luxS* strain displayed phagocytosis levels comparable to the STM WT (Fig 1B). To determine whether AI-2 signaling is required for intracellular survival rather than uptake,

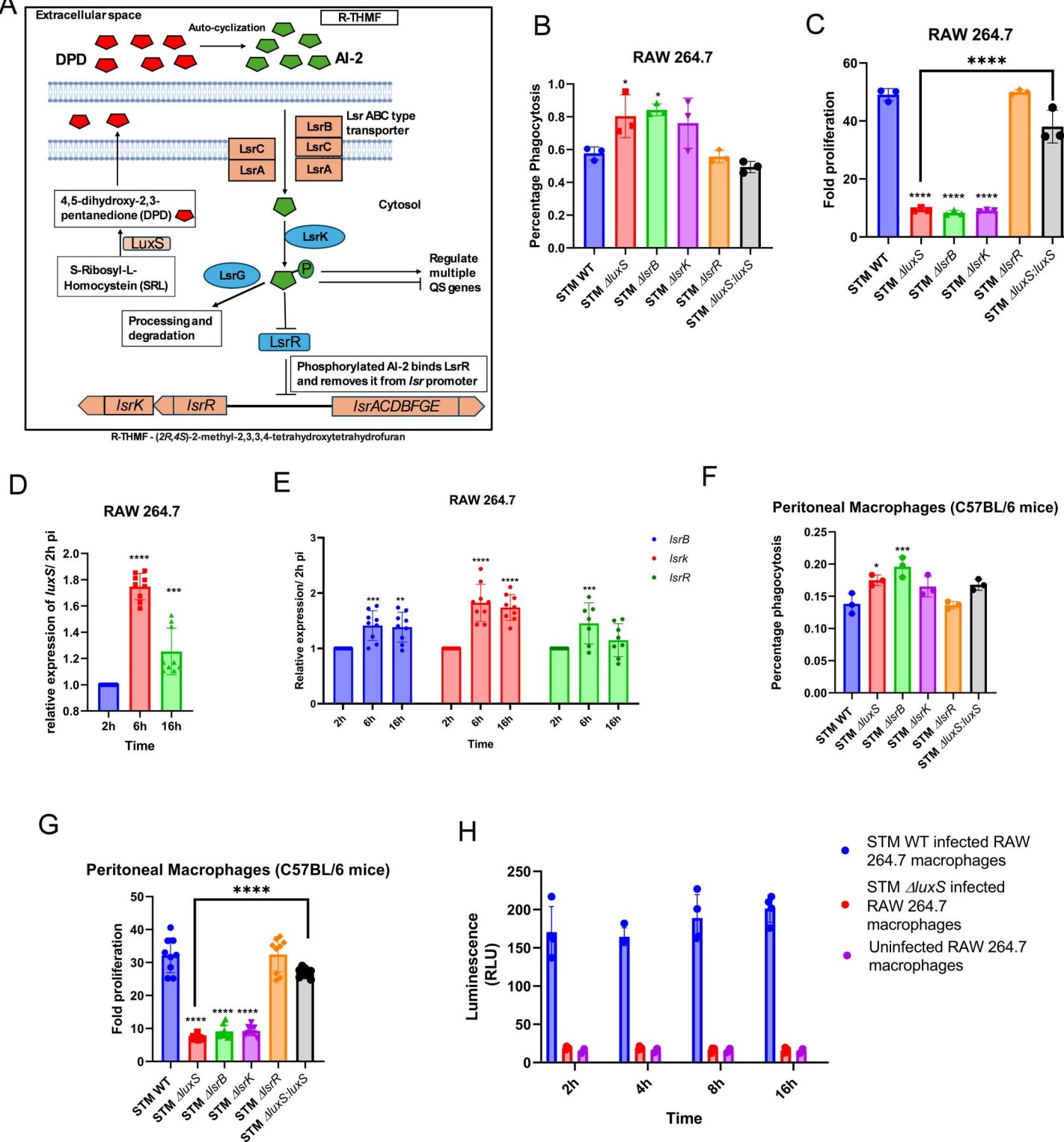

**Fig 1. LuxS/AI-2 quorum sensing is required by *Salmonella* Typhimurium to survive in the macrophages. (A)** Schematic of AI-2 synthesis and processing in *Salmonella* Typhimurium. **(B)** Percentage phagocytosis, and **(C)** Fold proliferation upon infection in RAW 264.7 macrophages (All data

in **(B)** and **(C)** are represented as mean±SD from three independent experiments). **(D)** *luxS* gene **(E)** *lsrB, lsrk,* and *lsrR* expression in STM WT upon infection in RAW264.7 macrophages (normalization relative to 2h) (data is from one experiment representative of 3 independent experiments). **(F)** Percentage phagocytosis and **(G)** Fold proliferation, upon infection in peritoneal macrophages (data is from one experiment representative of 3 independent experiments). **(H)** Autoinducer assay – AI-2 activity in RAW 264.7 macrophages upon infection (data is from one experiment representative of 3 independent experiments, n=4). One-way ANOVA with Dunnett's post-hoc test was used to analyze the data; Two-way ANOVA was used to analyze the grouped data; p values **** $p < 0.0001$, *** $p < 0.001$, ** $p < 0.01$, * $p < 0.05$.

we next evaluated bacterial proliferation within macrophages. Despite their increased phagocytosis, STM Δ*luxS*, STM Δ*lsrB*, and STM Δ*lsrK* showed impaired intracellular survival in RAW 264.7 cells (Fig 1C). By contrast, as expected, the *lsrR* mutant and the *luxS*-complemented STM Δ*luxS* strain behaved similarly to STM WT (Fig 1C). Our confocal microscopy data suggest that, under the tested conditions, STM Δ*luxS*, STM Δ*lsrB*, and STM Δ*lsrK* show reduced intracellular proliferation compared with STM WT (S1G and S1H Fig). Further, we infected RAW 264.7 macrophages with *Salmonella* strains for an intracellular gene expression study and observed that the *luxS* and the *lsrB, lsrK,* and *lsrR* genes were upregulated 1.5–2-fold from 6h post-infection to 16h compared to 2h post-infection (Fig 1D and 1E). We further corroborated our results by using primary macrophages derived from the peritoneal lavage of C57BL/6J mice for infection. The *luxS, lsrB,* and *lsrK* mutants also exhibited an increased uptake by the primary macrophages, but their proliferation was compromised (Fig 1F and 1G). Furthermore, we checked for intracellular AI-2 production by STM WT, STM Δ*luxS*, and STM Δ*lsrB* upon their infection of RAW 264.7 cells and observed the AI-2 production by STM WT and STM Δ*lsrB* till 16h post-infection, but not by STM Δ*luxS* (Fig 1H). Taken together, these findings imply that *Salmonella* Typhimurium requires LuxS/AI-2 signaling to survive in the hostile environment of macrophages.

## STM Δ*luxS* acquires AI-2 from STM WT during its *in vitro* growth

To find out if STM Δ*luxS* can use AI-2 molecules produced by STM WT, we co-infected RAW 264.7 macrophages with STM WT and STM Δ*luxS* (1:1 ratio), and observed that STM WT outcompeted STM Δ*luxS,* hinting that STM Δ*luxS* may not be able to utilize the AI-2 from STM WT within host macrophages (Fig 2A). The limitation of this experiment is that we cannot conclusively determine whether STM Δ*luxS* utilizes AI-2 from STM WT in the coinfection study in RAW 264.7 cells. Furthermore, the study does not confirm whether both bacterial strains infect the same cell, as observations reflect a global phenotype. However, it might be possible that STM Δ*luxS* cannot utilize AI-2 even when infecting the same cell due to a repressed *lsr* operon. On the contrary, when STM WT and STM Δ*luxS* were cocultured in LB medium before infection into macrophages, STM WT and STM Δ*luxS* proliferated similarly in RAW 264.7 cells (Fig 2B). To check if STM Δ*luxS* can utilize the AI-2 molecule produced by STM WT, we treated STM Δ*luxS* with the STM WT spent medium and exogenous DPD molecule. Upon infection into RAW 264.7 macrophages, the treated STM Δ*luxS* proliferated similarly to STM WT, suggesting its ability to use the AI-2 from the spent medium of STM WT or exogenous DPD (Figs 2C, 2D, and S2A–S2D). Next, we used a commercially available inhibitor (Z)-4 bromo-5-(bromomethyl)-3-methylfuran-2(5H) (BF-8) of LuxS/AI-2 signaling [24,25] at different concentrations with *Salmonella* Typhimurium and observed that while BF-8 did not affect *Salmonella* Typhimurium growth *in vitro*, but it attenuated *Salmonella* survival in RAW 264.7 macrophages, similar to that observed with STM Δ*luxS* (Figs 2E, 2F, and S2E). We also confirmed that exogenous DPD induces luminescence, whereas the inhibitor BF-8 inhibits luminescence. We observed that upon treatment of *Salmonella* with synthetic DPD or the inhibitor BF-8, AI-2 activity, measured as luminescence using an autoinducer assay, was increased or decreased, respectively (S2F and S2G Fig). To further validate the effects of synthetic DPD or BF-8, peritoneal macrophages were infected with the respective treated cultures. The data suggested that while exogenous DPD enhanced STM Δ*luxS* survival, treatment with the AI-2 inhibitor BF-8 significantly attenuated STM WT proliferation (S2H and S2I Fig). These findings imply that the *luxS* mutant can utilize exogenous AI-2 during its *in vitro* growth.

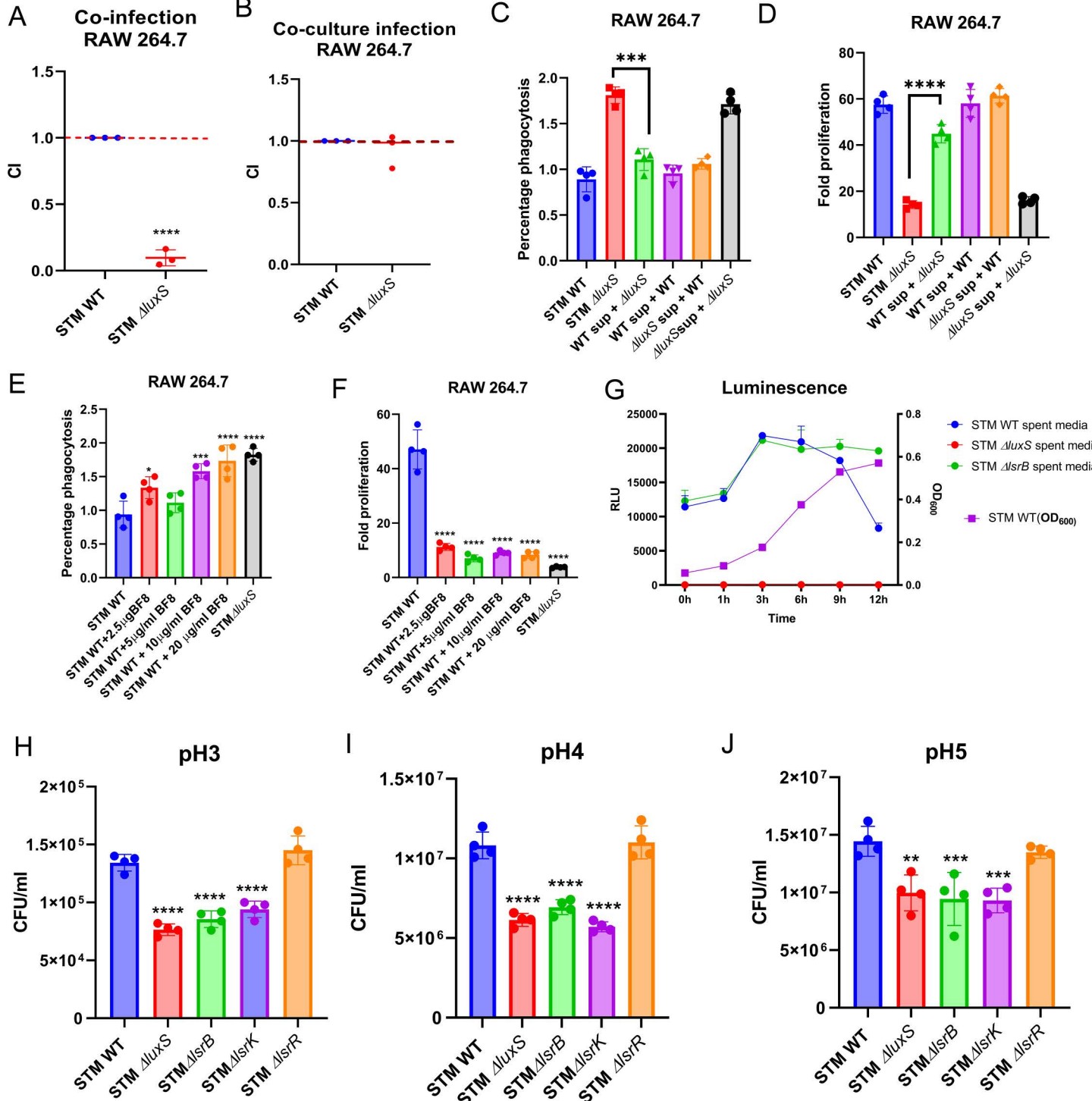

**Fig 2. STM Δ*luxS* acquires AI-2 from STM WT during its *in vitro* growth. (A)** Co-infection assay between STM WT and STM Δ*luxS upon* infection in RAW 264.7 macrophages at the ratio 1:1. **(B)** Co-culture grown STM WT and STM Δ*luxS* followed by infection in RAW 264.7 macrophages (All data in **(A)** and **(B)** are represented as mean ± SD from three independent experiments (N = 3, n = 3). **(C)** Percentage phagocytosis, **(D)** Fold proliferation of STM in RAW 264.7 macrophages upon 50% spent medium treatment of STM WT and STM Δ*luxS* (data is from one experiment representative of 3 independent experiments, N = 3, n = 4). **(E)** Percentage phagocytosis, **(F)** Fold proliferation of STM in RAW 264.7 macrophages upon treatment of BF-8 inhibitor

(data is from one experiment representative of 3 independent experiments, n = 4) **(G)** AI-2 production by STM WT, STM Δ*luxS*, and STM Δ*lsrB* in F-medium (pH 5) (Data is represented as mean ±SD of N = 3, n = 3). STM WT, STM Δ*luxS*, STM Δ*lsrB*, STM Δ*lsrK,* and STM Δ*lsrR* survival in different ranges of pH [3–8] of PBS **(H)** pH3 survival, **(I)** pH4 survival, **(J)** pH5 survival (data is from one experiment representative of 3 independent experiments, N = 3, n = 4). Unpaired Student's t-test was used to analyze the data; One-way ANOVA with Dunnett's post-hoc test was used to analyze the data; p values **** $p < 0.0001$, *** $p < 0.001$, ** $p < 0.01$, * $p < 0.05$.

## The loss of a functional AI-2 or its receptor-mediated sensing attenuates survival at acidic pH

The mechanisms underlying bacterial quorum sensing and pH sensing have been thoroughly investigated and shown to be distinct/independent gene regulatory systems. Our AI-2 bioassay in acidic medium shows AI-2 production by *Salmonella* (Fig 2G). Hence, we hypothesize that LuxS/AI-2 signaling may coordinate with the pH sensing system of *Salmonella* to regulate survival in an acidic environment. Firstly, when we assessed the expression of mRNAs of *luxS*, *lsrB*, *lsrK*, and *lsrR* during its growth in F-medium (mimics the acidic vacuolar condition), we observed ~1.5-fold increases in *luxS* at the mid-log phase to the late-log phase (S3A Fig), and *lsrB*, *lsrK,* and *lsrR* at the mid-log phase (S3B Fig). Moreover, we observed that *Salmonella* Typhimurium stably produces AI-2 even during its growth in F-medium (at pH 5), which is maximal during the mid to late-log phase. However, this reflects accumulated AI-2 rather than direct per-cell production (Fig 2G). The luminescence production continued in STM Δ*lsrB* spent medium even beyond 12 h (because there was no accumulation of AI-2 inside the STM Δ*lsrB*). However, there was no light production from the spent medium of STM Δ*luxS* (Fig 2G).

To underscore the role of AI-2 in facilitating survival in an acidic environment, we performed an acid survival assay in PBS and LB (a different range of pH) medium and noted that STM WT, STM Δ*luxS*, STM Δ*lsrB*, STM Δ*lsrK*, and STM Δ*lsrR* survived equally well at pH6 to pH8 in both medium (S3C and S3D Fig). However, at pH3, pH4, and pH5, STM WT survived significantly better compared to STM Δ*luxS*, STM Δ*lsrB*, STM Δ*lsrK* (Figs 2H–2J and S3E–S3G). We conclude that LuxS/AI-2 signaling and communication facilitate *Salmonella's* survival strategies in acidic environments under *in vitro* conditions.

## LuxS/AI-2 system tunes *phoP*/*phoQ* expression, thereby assisting in pH sensing and adaptation

A well-known two-component system (TCS) of PhoP/PhoQ is essential in mediating the acid tolerance response (ATR) in several Gram-negative bacteria, including *Salmonella* [26,27]. Thus, we conjectured that LuxS/AI-2 signaling modulates the *phoP*/*phoQ* gene expression, which, in turn, aids in the survival of *Salmonella* under an acidic environment. Thus, we questioned AI-2 signaling is essential to regulate the expression of the downstream gene targets apart from the *lsr* operon.

We first determined *phoP* gene expression in STM WT grown in LB (neutral), acidic LB (pH 5), F-medium, and during infection of RAW 264.7 macrophages (S4A–S4D Fig). *phoP* expression varies across conditions, with altered expression observed in LB (neutral and acidic), F-medium, and RAW macrophages, supporting the context-dependent regulation of *phoP* under environmental and intracellular conditions. We then compared *phoP* mRNA levels in STM Δ*luxS*, STM Δ*lsrB*, and STM Δ*lsrK* with STM WT and observed downregulation in LB, acidic LB (pH 5), and F-medium (Figs 3A, S4E, and S4F). Interestingly, the *phoP* expression was rescued when STM Δ*luxS* was grown in STM WT spent medium (neutral and acidic LB medium) (Figs 3B and S4G). Additionally, upon infection in macrophages, both *phoP* and *phoQ* were under-expressed in the mutants STM Δ*luxS*, STM Δ*lsrB*, and STM Δ*lsrK* compared to STM WT (Fig 3C and 3D). Hence, we conclude that AI-2 sensing and the signaling cascade orchestrate the *phoP*/*phoQ* expression in *Salmonella* Typhimurium.

Next, we evaluated the acid tolerance response (ATR) and noted that with a prior adaptation to pH 5, STM WT and STM Δ*lsrR* were resilient and showed better survival upon exposure to pH 3 compared to STM Δ*luxS,* STM Δ*lsrB*, and STM Δ*lsrK* (Fig 3E). As the STM Δ*luxS*, STM Δ*lsrB*, and STM Δ*lsrK* strains exhibited a diminished ATR, we determined the relative cytoplasmic acidification of all the strains using a pH-sensitive pHuji plasmid. We observed that all *Salmonella* strains maintained a neutral cytosolic pH in LB medium. However, when grown in acidic F-medium (pH 5), STM WT and

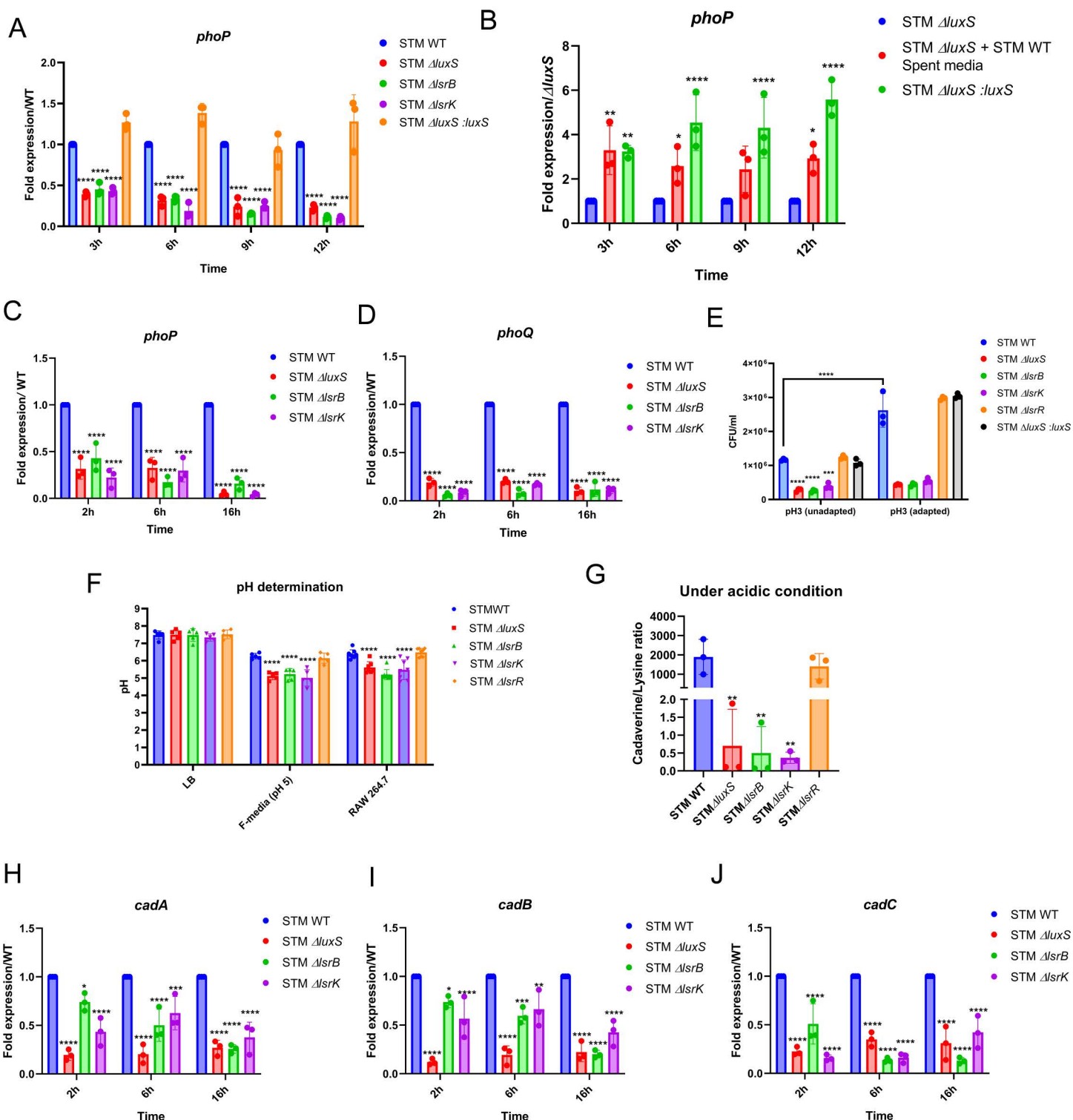

**Fig 3. LuxS/AI-2 system tunes *phoP* expression, thereby assisting in pH sensing and adaptation. (A)** *phoP* gene expression in STM WT, STM Δ*luxS*, STM Δ*lsrB*, and STM Δ*lsrK in vitro* conditions, LB medium (pH5). **(B)** mRNA expression of *phoP* in STM Δ*luxS* upon treatment of STM WT spent medium (LB, acidic pH 5). **(C)** *phoP* **(D)** *phoQ* expression in STM WT, STM Δ*luxS*, STM Δ*lsrB*, and STM Δ*lsrK* upon infection in RAW 264.7 macrophages. **(E)** STM WT, STM Δ*luxS*, STM Δ*lsrB*, STM Δ*lsrK*, and STM Δ*lsrR* acid adaptation and tolerance. **(F)** Cytosolic pH determination of STM WT,

STM Δ*luxS*, STM Δ*lsrB*, STM Δ*lsrK*, and STM Δ*lsrR* by phuji plasmid in LB medium, in F-medium (pH 5), and upon infection in RAW 264.7 macrophages. **(G)** Ratio of cadaverine and lysine under acidic conditions (pH 5) **(H)** *cadA*, **(I)** *cadB,* and **(J)** *cadC*, mRNA expression upon infection in RAW 264.7 macrophages. One-way ANOVA with Dunnett's post-hoc test was used to analyze the data; Two-way ANOVA was used to analyze the grouped data; p values **** $p < 0.0001$, *** $p < 0.001$, ** $p < 0.01$, * $p < 0.05$. All data are represented as mean ± SD from three independent experiments (N = 3, n ≥ 3).

STM Δ*lsrR* could maintain their cytosolic pH within the near-neutral range, while cytosolic pH for STM Δ*luxS*, STM Δ*lsrB*, and STM Δ*lsrK* dropped to an acidic range and a comparable reduction was observed within the hostile macrophage environment (Figs 3F and S4H–S4L). This indicates that these mutants are less able to maintain intracellular pH homeostasis under acidic stress.

The PhoP/PhoQ TCS aids in pH homeostasis via the *cadC/BA* system (transcriptional regulator, lysine/cadaverine antiporter, and lysine decarboxylase), thereby promoting bacterial survival in mildly acidic environments [28,29]. So, we measured the expression of *cadC*, *cadB*, and *cadA* in STM WT and STM Δ*phoP* upon infection in RAW 264.7 macrophages and observed diminished mRNA expression of *cadC*, *cadB*, and *cadA* in STM Δ*phoP* upon infection in RAW 264.7 macrophages, validating the above observation (S5A–S5C Fig). The lysine decarboxylase (CadA) quenches the $H^+$ while converting lysine to cadaverine, which is exported to the extracellular milieu by CadB [28]. We performed mass spectrometry analysis to quantify cadaverine and lysine in the culture supernatant. Here, we noted the cadaverine/lysine ratio to be low in STM Δ*luxS*, STM Δ*lsrB*, and STM Δ*lsrK* compared to STM WT under an acidic pH of 5 (Fig 3G). Additionally, in the mutants, the *cadA*, *cadB*, and *cadC* genes were under-expressed at 2h, 6h, and 16h post-infection into RAW 264.7 cells (Fig 3H–3J). From a mechanistic perspective, we conclude that LuxS/AI-2 signaling regulates ATR and maintains the cytosolic pH of *Salmonella* by regulating the *cadC/AB* operon via *phoP/phoQ*.

## LuxS/AI-2 signaling regulates *phoP/phoQ* expression through LsrR interaction with the *phoP* promoter

We explored whether LsrR, the downstream regulator of the LuxS/AI-2 signaling pathway, could directly interact with the *phoP* promoter to influence its activity. The AlphaFold3 structure suggested that LsrR could bind to both the DNA strands of the *lsr* promoter, but to only one of the strands of the *phoP* promoter [30]. The predicted interactions are mediated by residues Y25, T31, Q32, S33, R43, L44, K45, S47, and R48, all within a distance of <5 Å and a predicted aligned error (PAE) of <14 across multiple replicates (Figs 4A, 4B, S6A, and S6B). These residues are located in the N-terminal domain, spanning positions 25–48, between alpha helices 1, 2, and 3. Notably, Y25, T31, and R43 showed the most significant interactions, with R43 from helix 3 displaying the highest number of contact points [31]. A covariance analysis of the multiple sequence alignment revealed that R43 most frequently co-occurs with Y25, an interacting residue located in helix 1, with the highest probability among all analyzed residues [32]. Given the strong predicted interactions of R43 and Y25 with the *lsr* and *phoP* promoters, we generated Y25A, R43A, and Y25A/R43A mutants and used AlphaFold3 to predict structural impacts. The AlphaFold structures indicated that the mutations disrupted promoter interactions, with PAE values exceeding 18 and pLDDT scores for all the DNA-contacting residues dropping below 30. Additionally, we performed NPT ensemble molecular dynamics simulations for the mutant protein-DNA complex, revealing a significant weakening of interactions. We validated our findings *in vitro* through electrophoretic mobility shift assays (EMSA) using purified wild-type LsrR and its Y25A, R43A, and Y25A/R43A mutants to assess their binding to HPLC-purified *lsr* and *phoP* promoters (Figs 4C, 4D, and S6C–S6F). We observed that WT LsrR binds to both the promoter, whereas the double mutant loses binding. Further, R43A mutant LsrR reduced it, and the double mutant lost nearly all binding at 200mM NaCl (S7A and S7B Fig). A similar result was observed for the interaction of LsrR with a single-stranded promoter sequence (S7C–S7F Fig), and a random DNA sequence did not show any interaction (S7G Fig). So, LsrR is predicted, based on *in-silico* analysis, to preferentially interact with the single-stranded *phoP* promoter; however, EMSA results show that it binds to both single- and double-stranded forms of the *phoP*/Q promoter. To assess regulatory effects during infection, we generated a *luxS*/*lsrR* double knockout and complemented it with *lsrR* (WT) and a mutant *lsrR gene*. Next, we performed

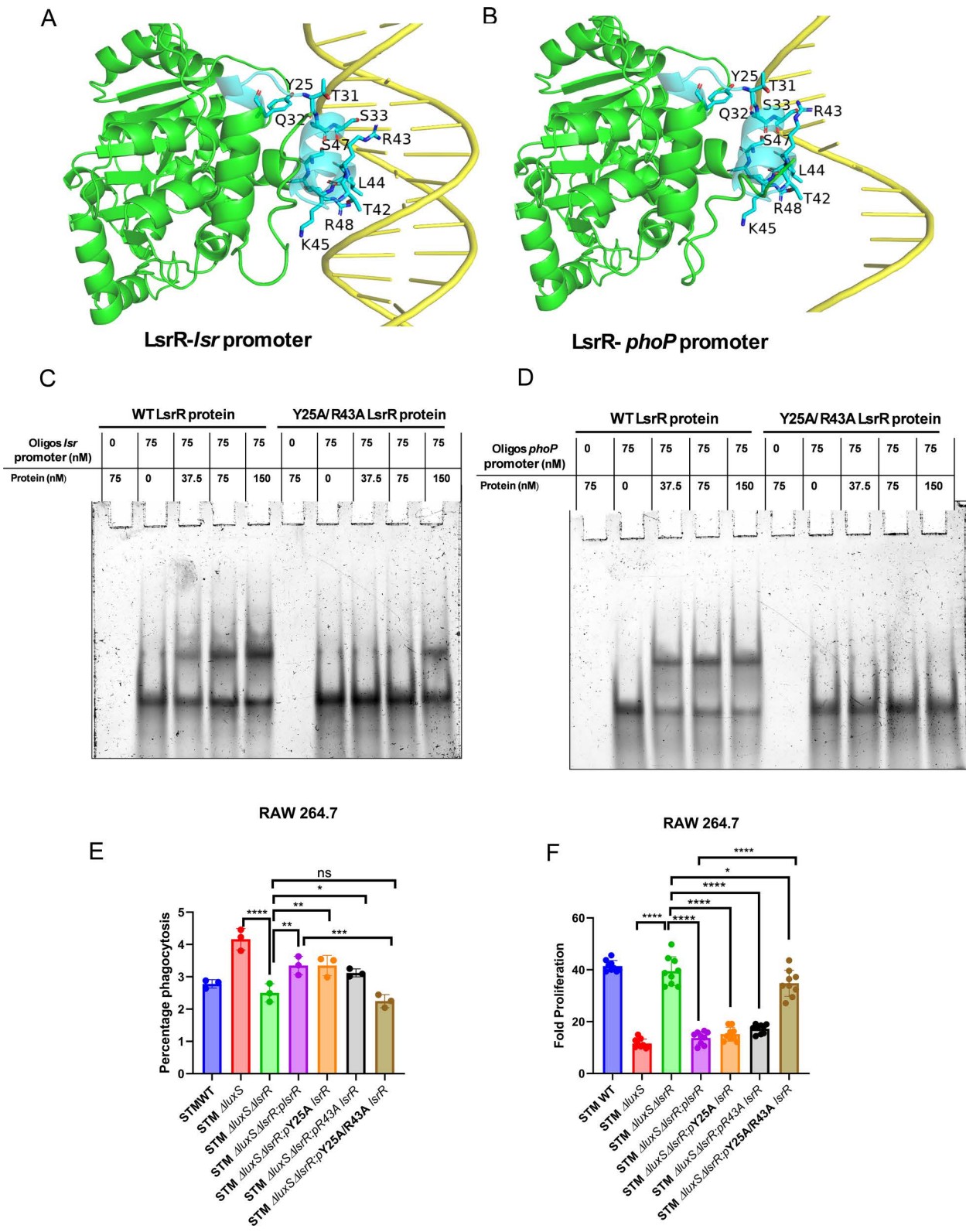

**Fig 4. LuxS/AI-2 signaling regulates *phoP/phoQ* expression through LsrR interaction with the *phoP* promoter in *Salmonella*.** AlphaFold3 structural models visualised using PyMol depicting LsrR (green) interactions with the *phoP* and *lsr* promoters (yellow). The N-terminal domain residues,

spanning helices 1, 2, and 3, are highlighted in blue, and the key interacting residues of LsrR are represented with a ball and stick model **(A, B)**. Electrophoretic mobility shift assays (EMSA) assessing double-strand DNA binding of wild-type LsrR and its Y25A/R43A mutant to *lsr* and *phoP* promoters, respectively. (data is from one experiment representative of 3 independent experiments) **(C, D)**. Intracellular survival assay (ICSA) with mutated Y25A, R43A, and Y25A/R43A *lsrR* cloned in pQE-60 complemented strains **(E)** phagocytosis and **(F)** Fold proliferation (data is from one experiment representative of 3 independent experiments). One-way ANOVA with Dunnet's post-hoc test was used to analyze the data; p values **** $p < 0.0001$, *** $p < 0.001$, ** $p < 0.01$, * $p < 0.05$.

infection assays in RAW 264.7 macrophages using the *luxS*/*lsrR* double knockout strain complemented with either WT *lsrR* or mutant *lsrR*. While the double knockout strain proliferated similarly to STM WT, *lsrR* complementation reduced proliferation. The Y25A/R43A mutant restored survival to double knockout levels (Fig 4E and 4F). However, we further infected RAW264.7 macrophages with *phoP* complemented under a non-native promoter in STM Δ*luxS* and observed rescued survival of STM Δ*luxS* (S8A and S8B Fig). Together, these data indicate that LsrR binding to the *lsr* and *phoP* promoters acts as a negative regulator of *Salmonella* pathogenicity.

## AI-2 orchestrates the *Salmonella* Pathogenicity Island-2 gene expression by sensing low pH

PhoP has been shown to control the TCS SsrA/SsrB at the transcription level, which are the master regulators of the SPI-2 genes [33]. The expression of *ssrA* and *ssrB* genes was also downregulated in STM Δ*phoP* upon infection into RAW 264.7 cells (S9A and S9B Fig). Thus, we hypothesized that LuxS/AI-2 signaling might control the SsrB/SsrA system via PhoP/PhoQ. Upon infection into RAW 264.7 macrophages, the mRNA expression of *ssrB* was downregulated from early stage post-infection to late stage post-infection in macrophages (Fig 5A). On the other hand, *ssrA* was downregulated at a late stage post-infection (Fig 5B). Furthermore, we observed that expression of *ssaV*, a Type 3 secretion system (T3SS) needle complex protein encoded by SPI-2, and *spiC*, an SPI-2-encoded effector molecule, was under-expressed in the absence of *luxS*, *lsrB*, and *lsrK* genes in *Salmonella* upon infection of RAW 264.7 macrophages (Fig 5C and 5D). Since SPI-2 effectors are known to modulate host immune signaling and promote an anti-inflammatory (M2-like) phenotype favorable for intracellular survival, we hypothesized that quorum-sensing–defective mutants would impair this modulation [34,35]. Notably, this impaired SPI-2 gene expression correlated with altered host cytokine responses (Proinflammatory *Tnf-α* and anti-inflammatory *Il-10*) wherein STM WT promoted a more anti-inflammatory (*Il-10*, M2-like) macrophage profile compared to quorum-sensing–defective mutants (S9C and S9D Fig).

So far, our findings convey that the LuxS/AI-2 signaling regulates *phoP*/*phoQ* genes and, thereby, the SPI-2 cluster in *Salmonella*. However, whether AI-2 directly or the AI-2-PhoP/PhoQ axis controls the SPI-2 genes remains unclear. Thus, we cloned the *phoP*/*phoQ* operon under a non-native promoter in the pQE60 vector. Firstly, we confirmed the *phoP* mRNA overexpression level in STM WT: pQE60-*phoP*/*phoQ* and STM Δ*luxS*: pQE60-*phoP*/*phoQ* upon infection in RAW 264.7 (S9E and S9F Fig). Using these complemented *Salmonella* strains, we observed that STM Δ*luxS*, STM Δ*lsrB*, and STM Δ*lsrK* with a *phoP*/*phoQ* cloned under a non-native promoter survived equally at acidic pH3, 4, and 5 as STM WT (Fig 5E–5G). Furthermore, the STM Δ*luxS*, STM Δ*lsrB*, and STM Δ*lsrK* with cloned *phoP*/*phoQ* showed enhanced ATR response (Fig 5H). Thus, when *phoP*/*phoQ* is not under its native promoter, the regulation by LuxS/AI-2 is not required. Also, *ssrB* and *ssrA* expression are significantly higher in *phoP*/*phoQ* complemented knockout strains compared to STM Δ*luxS*, STM Δ*lsrB*, and STM Δ*lsrK* strains (Figs 5I–5K and S9G–S9I). Cumulatively, these results suggest that, indeed, the LuxS/AI-2 signalling controls ATR by regulating the *phoP*/*phoQ* genes.

## The absence of AI-2 mediated signaling compromises the *in vivo* colonization of *Salmonella* in mice

Because AI-2 is a universal signaling molecule, it is an excellent choice for mediating interactions between cells in the mammalian gut, where hundreds of bacterial species live and interact. Thus, we aimed to understand the role of LuxS/AI-2 signalling in regulating *Salmonella* pathogenesis in *in vivo* mouse models. We infected C57BL/6J mice by orally

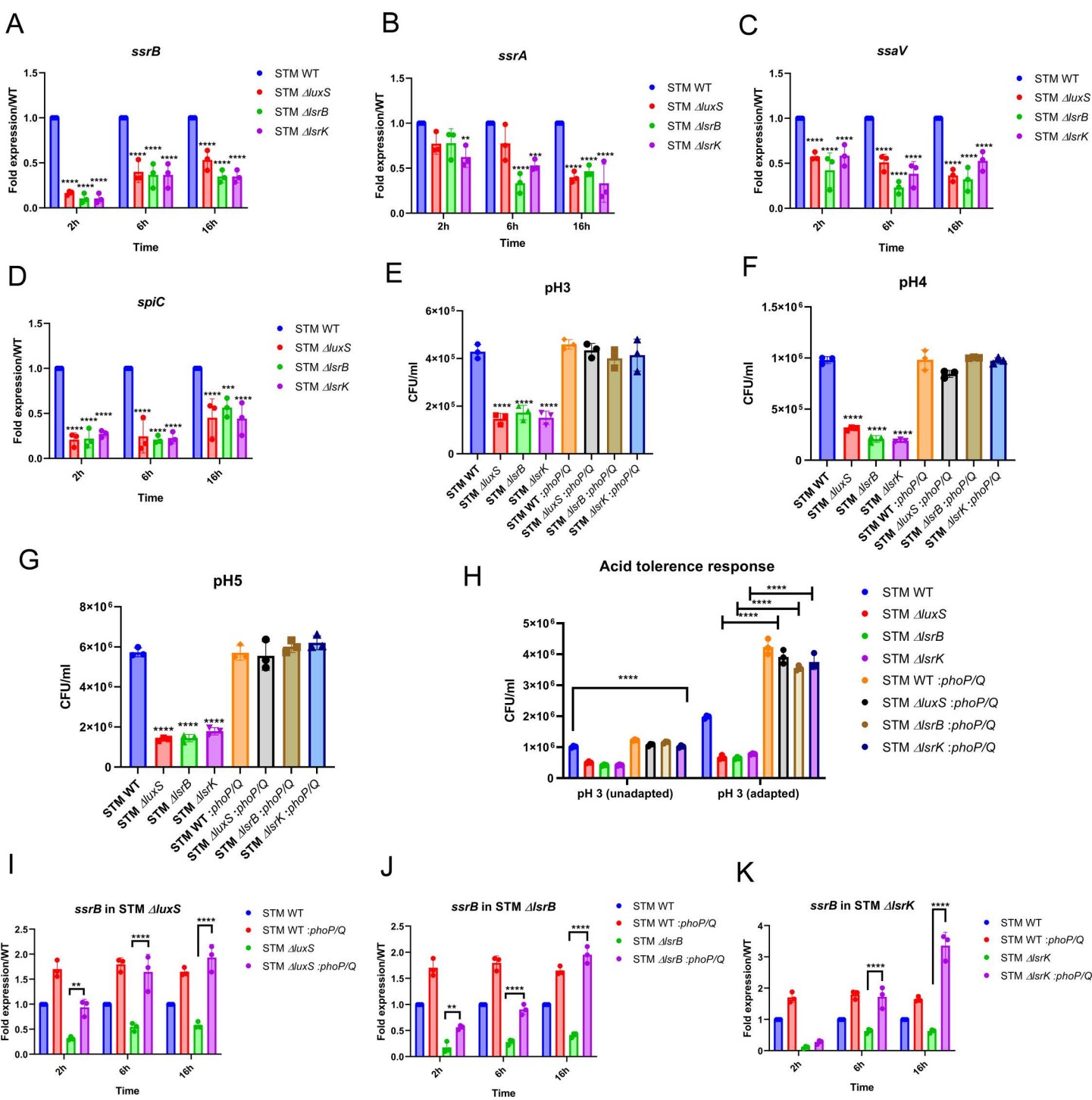

**Fig 5. AI-2 orchestrates the *Salmonella* pathogenicity island-2 gene expression by sensing low pH. (A)** *ssrB* **(B)** *ssrA* **(C)** *ssaV* **(D)** *spiC,* mRNA expression in STM WT, STM Δ*luxS*, STM Δ*lsrB*, and STM Δ*lsrK* upon infection in RAW 264.7 macrophages. STM WT, STM Δ*luxS*, STM Δ*lsrB*, STM Δ*lsrK,* STM WT: *phoP*, STM Δ*luxS*: *phoP*, STM Δ*lsrB*: *phoP*, STM Δ*lsrK*: *phoP* survival in PBS at **(E)** pH 3, **(F)** pH 4, **(G)** pH 5. **(H)** STM WT, STM Δ*luxS*, STM Δ*lsrB*, STM Δ*lsrK,* STM WT:*phoP*, STM Δ*luxS*:*phoP*, STM ΔlsrB:*phoP*, STM Δ*lsrK*: *phoP* acid adaptation and tolerance. *ssrB* mRNA expression in *phoP/phoQ* cloned strain of **(I)** STM Δ*luxS,* **(J)** STM Δ*lsrB* and **(K)** STM Δ*lsrK.* One-way ANOVA with Dunnet's post-hoc test was used to analyze the data; Two-way ANOVA was used to analyze the grouped data; p values **** p < 0.0001, *** p < 0.001, ** p < 0.01, * p < 0.05. All data are represented as mean ± SD from three independent experiments (N = 3, n = 3).

gavaging with STM WT and knockout strains at a CFU of $10^7$ per mouse (Fig 6A). We observed that STM Δ*luxS*, STM Δ*lsrB*, and STM Δ*lsrK* showed a significantly lower organ burden in the intestine, mesenteric lymph node (MLN), spleen, and liver, and lower bacteraemia upon oral gavage (Fig 6B–6F). Oral gavage mimics the physiological route of *Salmonella* infection into its host, and *Salmonella* is required to be able to breach the intestinal barrier successfully. The lower organ colonization in STM Δ*luxS*, STM Δ*lsrB*, and STM Δ*lsrK* can be explained by their inability to cross the gut epithelial barrier. Next, we bypassed the gut epithelial barrier by infecting C57BL/6 mice intraperitoneally and found that STM Δ*luxS,* STM Δ*lsrB, and* STM Δ*lsrK* still exhibited reduced colonization in the spleen and liver and less dissemination in blood than STM WT (S10A–S10D Fig). Furthermore, upon infection of mice by oral gavage at a CFU of $10^8$ per mouse, we noted that the mice infected with STM WT and STM Δ*lsrR* succumbed to death as early as the 6th day of post-infection, while STM Δ*luxS*, STM Δ*lsrB*, and STM Δ*lsrK* infected mice survived longer (Fig 6G–6H) with less weight reduction (S10E Fig). Further, we observed that the BF-8 inhibitor at 4 mg/kg reduced the STM WT colonization to different sites of infection in C57BL/6 mice (Fig 6I–6N) and improved the survival of mice with delayed weight reduction compared to untreated mice (Figs 6O, 6P, and S10F). Liver tissue histopathology results also suggest that treatment with BF-8 reduced the disease score (S9G Fig). Finally, we confirmed our observation by using *phoP* complemented under the non-native promoter STM Δ*luxS.* We infected C57BL/6J mice by orally gavaging strains at a CFU of $10^7$ per mouse. Post 5th day of infection, we determined the organ burden and observed that STM Δ*luxS:* pQE60-*phoP*/*phoQ* (*phoP* complemented under non-native promoter) colonized significantly higher in the intestine, MLN, Liver, Spleen, and more disseminated in blood compared to STM Δ*luxS* (S11A–S11E Fig). Furthermore, upon infection of mice by oral gavage at a CFU of $10^8$ per mouse, we noted that the mice infected with STM Δ*luxS:* pQE60-*phoP*/*phoQ* succumbed to death earlier compared to STM Δ*luxS,* infected (S11F and S11G Fig). Conclusively, our results show that LuxS/AI-2 signaling is critical for the *in vivo* pathogenesis and virulence of *Salmonella* Typhimurium.

## Discussion

The human body serves as a host for a vast array of microorganisms collectively known as the normal microbiota. These microbes coexist closely with their host and often offer significant health benefits, especially within the intestine. Given the multispecies nature of this environment, cross-species signaling likely enables microbes to fine-tune their behaviors in response to social and environmental cues. These processes are often coordinated through quorum sensing, a communication system in which bacteria release chemical signals called autoinducers to monitor population density and regulate gene expression collectively [37]. Studies show that AI-2 facilitates interspecies communication and modulates behavior across species lines, leading to the hypothesis that AI-2 plays a central role in regulating microbial interactions and community dynamics in the gut [38].

The aspects of the bacterial lifestyle and pathogenesis rely on the bacterial quorum and community attributes, like for *Salmonella*, the infectious dose to cause the illness in humans is more than 10,000 CFU [39]. Autoinducers are chemical signal molecules that underpin bacterial communication and differ amongst species, just like the different languages that exist across the human civilization. Here, we report that as a successful enteric pathogen, *Salmonella* can use the AI-2 molecule to coordinate and break the colonization resistance in the gut. AI-2 synthesis and *lsr* operon are concurrently regulated under neutral and acidic conditions in *Salmonella,* aiding in its survival. The phosphorylation of AI-2 by LsrK is crucial for its function [40] and is supported by our study, which shows that deletion of *lsrK* renders *Salmonella* less able to survive in macrophages and to upregulate *phoP* expression*.*

The gastrointestinal tract presents with a dynamic environment where microbes compete for space and nutrients through various mechanisms. Many gut-associated bacteria encode LuxS or produce AI-2, including a significant proportion of Firmicutes and Proteobacteria, and some species of Bacteroidetes and Actinobacteria [41]. Enteric pathogens like *Salmonella* might benefit from the AI-2 produced by the gut commensals in the intestinal lumen. Our *in vivo* studies justify it to some extent with compromised colonization of the primary and secondary sites of infection upon abrogation of LuxS/

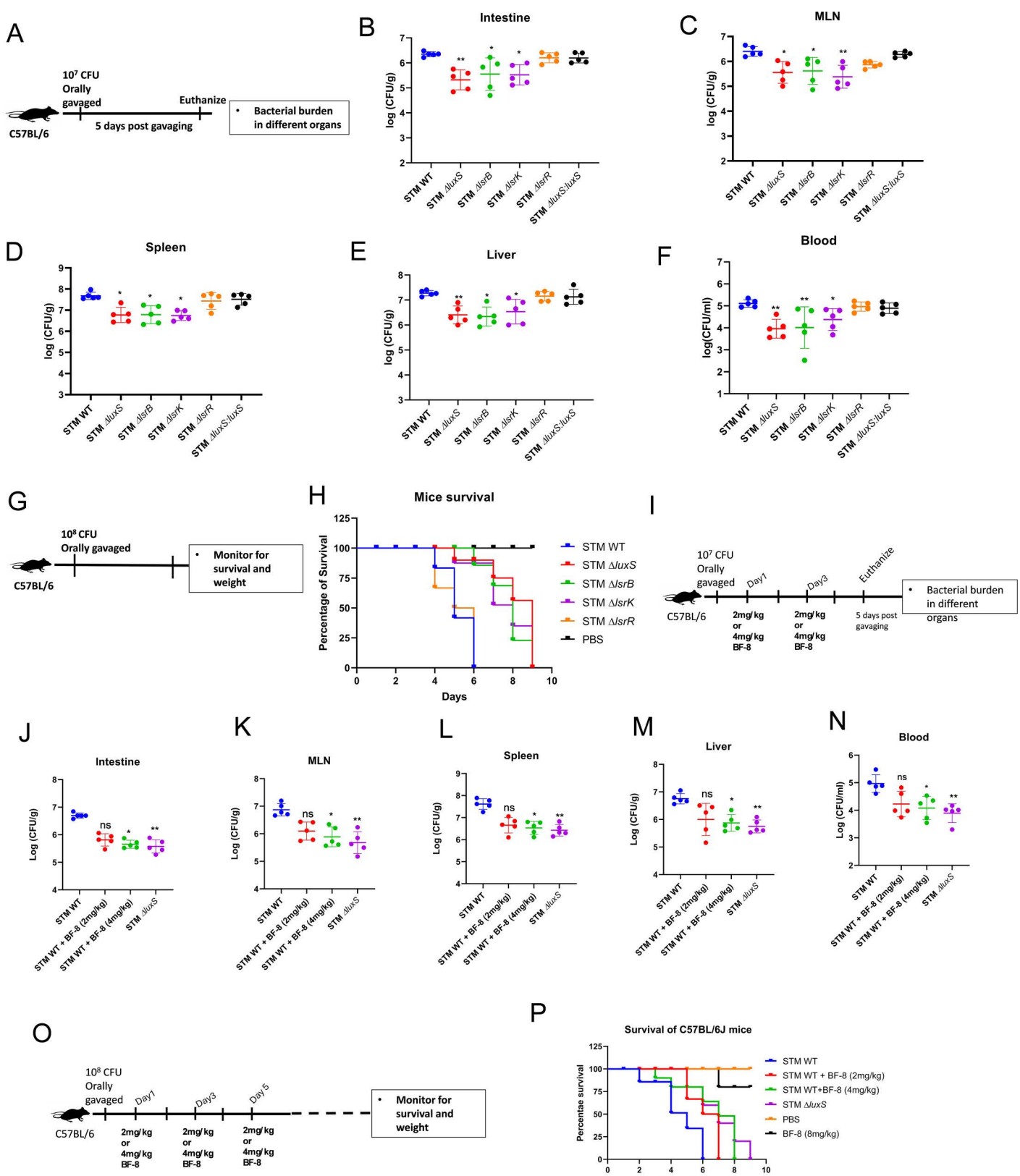

**Fig 6. The absence of AI-2 mediated signaling compromises the *in vivo* colonization of *Salmonella* in mice. (A)** The experimental protocol for organ burden in C57BL/6J mice by orally gavaging $10^7$ CFU per mouse [36]. The organ burden post 5 days of oral gavage in **(B)** Intestine, **(C)**Mesenteric

lymph node (MLN), **(D)** Spleen, **(E)** Liver, and dissemination in **(F)** Blood. **(G)**. The experimental protocol for C57BL/6J mice survival by orally gavaging $10^8$ CFU of STM per mouse. **(H)** Percentage survival of Mice upon infection with STM WT, STM Δ*luxS*, STM Δ*lsrB*, STM Δ*lsrK,* STM Δ*lsrR,* and STM Δ*luxS:luxS.* **(I)** The experimental protocol for organ burden in C57BL/6J mice by orally gavaging $10^7$ CFU per mouse upon treatment of BF-8 inhibitor (2mg/kg and 4mg/kg through the Intraperitoneal route). The organ burden post 5 days of oral gavage in **(J)** Intestine, **(K)** Mesenteric lymph node (MLN), **(L)** Spleen, **(M)** Liver, and blood dissemination in **(N)** Blood. **(O)** The experimental protocol of mice survival in C57BL/6J mice by orally gavaging $10^8$ CFU per mouse upon treatment of BF-8 inhibitor (2mg/kg and 4mg/kg through the Intraperitoneal route). **(P)** Percentage survival of Mice upon infection with STM WT, STM WT upon treatment, and STM Δ*luxS.* Non-parametric One-way ANOVA (Kruskal-Wallis) with Dunn's post hoc test was used to analyze organ burden in mice. (data is from one experiment representative of 2 independent experiments, N = 2, n = 5).

**AI-2 signaling in *Salmonella*.** Although the neighboring bacteria might produce AI-2 in the intestinal niche, STM Δ*lsrB* and STM Δ*lsrK* cannot utilize AI-2 to mediate the downstream signaling cascades. Thus, we underscore the critical function of individual members of this cascade, the AI-2 synthesizing enzyme, receptor, and kinase enzyme. Our study of bacterial strain co-culture prior to infection further suggests that *Salmonella* can utilise the AI-2 molecules produced by the gut microbiota.

During intestinal colonization, a prime strategy of this pathogen is to activate the ATR systems to endure the harsh environment. The four major regulators that control relevant stress responses in *Salmonella* are RpoS, PhoPQ, Fur, and OmpR/EnvZ [42]. We describe a mechanism that AI-2 signaling tunes the expression of the two-component system *phoP/Q*, which helps in ATR. Relating to intravacuolar survival, studies show that the acidic pH of the vacuole is necessary for the assembly of T3SS encoded by SPI-2, but not for the effector molecule's secretion [43]. Instead, secretion is induced upon exposure to a neutral pH environment, such as contact of the T3SS2 needle with the host cell cytosol or a shift from acidic to neutral pH conditions [44].

With respect to macrophage polarization, although we did not explicitly classify macrophages into defined M1 or M2 phenotypes, we assessed host inflammatory responses by measuring key cytokines following infection. The observed differences in pro- and anti-inflammatory cytokine expression, including TNF-α and IL-10, suggest that STM WT more effectively promotes an anti-inflammatory, M2-like state. In contrast, quorum-sensing–defective mutants exhibit an altered cytokine profile, indicative of impaired immune modulation. These findings are consistent with their reduced intracellular survival and diminished SPI-2 activation and collectively provide functional insight into the macrophage activation states during infection. In this context, our findings suggest that disruption of AI-2 signaling impairs the ability of *Salmonella* to cope with acidic stress, which may indirectly compromise proper SPI-2 function and effector secretion during intracellular infection. Thus, AI-2 signaling appears to contribute to bacterial fitness in acidic environments, thereby supporting optimal SPI-2 dependent survival and proliferation within macrophages. Moreover, *Salmonella* AI-2 phosphorylation by LsrK is a key process in controlling the downstream targets. As noted earlier, a limitation of this study is that we cannot determine whether STM Δ*luxS* utilizes AI-2 produced by STM WT during coinfection in RAW 264.7 macrophages. In addition, our approach does not resolve whether both strains infect the same host cell, as the observations reflect a population-level phenotype. Future studies employing single-cell resolution approaches will be required to resolve these possibilities and further delineate AI-2 mediated interactions during infection.

We broaden the horizon of LsrR regulation of gene expression beyond the *lsr* operon by unraveling the binding of LsrR to the *phoP* promoter via its Y25 and R43. Thijs et al. concluded that the LsrR regulon comprises genes directly regulated by LsrR through canonical promoter binding, as identified by genome-wide transcriptional profiling, and did not find the *phoP* promoter among LsrR-regulated targets [45]. In contrast, our findings predicted that LsrR regulates *phoP* via interaction with single-stranded promoter DNA rather than canonical double-stranded DNA binding. Interestingly, these insights also shed light on an unusual interaction of the LsrR with a single strand

of the DNA at the promoter site. This not only identifies the mechanism but also paves the way to a "secret garden" of genes that may be regulated by LsrR. However, our EMSA results show that LsrR binds both single- and double-stranded forms of the *phoP/Q* promoter *in vitro*, indicating no strict structural specificity under these conditions. But *in silico* analysis suggests a stronger and more specific interaction with the single-stranded form. This apparent discrepancy likely arises from fundamental differences between the two approaches. EMSA assays are performed under simplified conditions and therefore primarily assess binding capability, but do not capture the dynamic structural states of DNA in the cellular environment. *In vivo*, DNA undergoes processes such as transcription, supercoiling, and local unwinding, which can generate transient single-stranded regions that may preferentially recruit LsrR. Thus, while EMSA demonstrates that LsrR can bind both single- and double-stranded DNA, the *in silico* prediction may reflect a context-dependent preference for single-stranded regions under physiological conditions. Further *in vivo* and structure-specific studies will be required to determine the biological relevance of this interaction. While this observation is in *Salmonella,* it can be extrapolated to other bacteria that utilise a similar AI-2 signalling cascade. Overall, we identify a regulatory network via AI-2/LsrR and PhoP/PhoQ TCS, leading to a remarkable survival strategy under acidic pH. A proposed strategy for controlling bacterial infections focuses not on bactericidal activity, as in the case of conventional antimicrobials, but rather on disrupting a key functional process critical to the pathogen's infectious lifecycle. One promising target is the communication, wherein *Salmonella* uses it to actively survive in the acidic environment of the stomach, colonizes in the intestine, followed by a secondary site of infection, and survives in the hostile niche of macrophages, which is an essential step for successful pathogenesis. These virulence process depends on the coordinated regulation of numerous virulence genes, which are modulated through an intricate network of genetic and environmental signals. Our *in-vivo* study suggests that inhibition of LuxS/AI-2 signaling attenuates *Salmonella* colonization. Overall, we identify a novel regulatory network via AI-2 and PhoP/PhoQ TCS, leading to an astonishing survival strategy for *Salmonella* under acidic pH (Fig 7). Our findings highlight and open promising avenues to target bacterial communication and AI-2 signaling, for designing antibacterial strategies to limit bacterial infection.

## Materials and methods

### Ethics statement

All experiments comply with the rules set forth by the Indian Institute of Science, Bangalore's Institutional Animal Ethics Committee (IAEC). The approved protocol numbers are CAF/Ethics/852/2021 and CAF/Ethics/116/2025. The Committee for Control and Supervision of Experiments on Animals (CPCSEA), a statutory committee established under Chapter 4, Section 15(1) of the Prevention of Cruelty to Animals Act 1960, and National Animal Care provided guidelines that were meticulously adhered to during all animal experiments, all of which were approved by the Institutional Animal Ethics Committee. (Registration No. 435 48/1999/CPCSEA). The Institutional Animal Ethics Committee approved every animal experiment, and the National Animal Care Guidelines were meticulously followed.

### Bacterial strains and growth conditions

The wild-type *Salmonella enterica* serovar Typhimurium strain 14028S (STM WT) used in all experiments in this study was a kind gift from Professor Michael Hensel of the Max Von Pettenkofer-Institute for Hygiene und Medizinische Mikrobiologie in Germany. The bacterial strains were revived on LB agar with or without antibiotics. The LB broth culture of wild-type, knockout, and complemented strains was cultured at 37˚C (170 rpm) in an orbital shaker incubator (S1 Table). Antibiotics Kanamycin (50 µg/ml), Chloramphenicol (25 µg/ml), and Ampicillin (50 µg/ml) were used whenever required. *Vibrio campbellii* ATTC BAA-1117 strain was used for autoinducer (AI-2) bioassay. This strain was grown in ATCC Medium: 2746 Autoinducer Bioassay (AB) Medium [47].

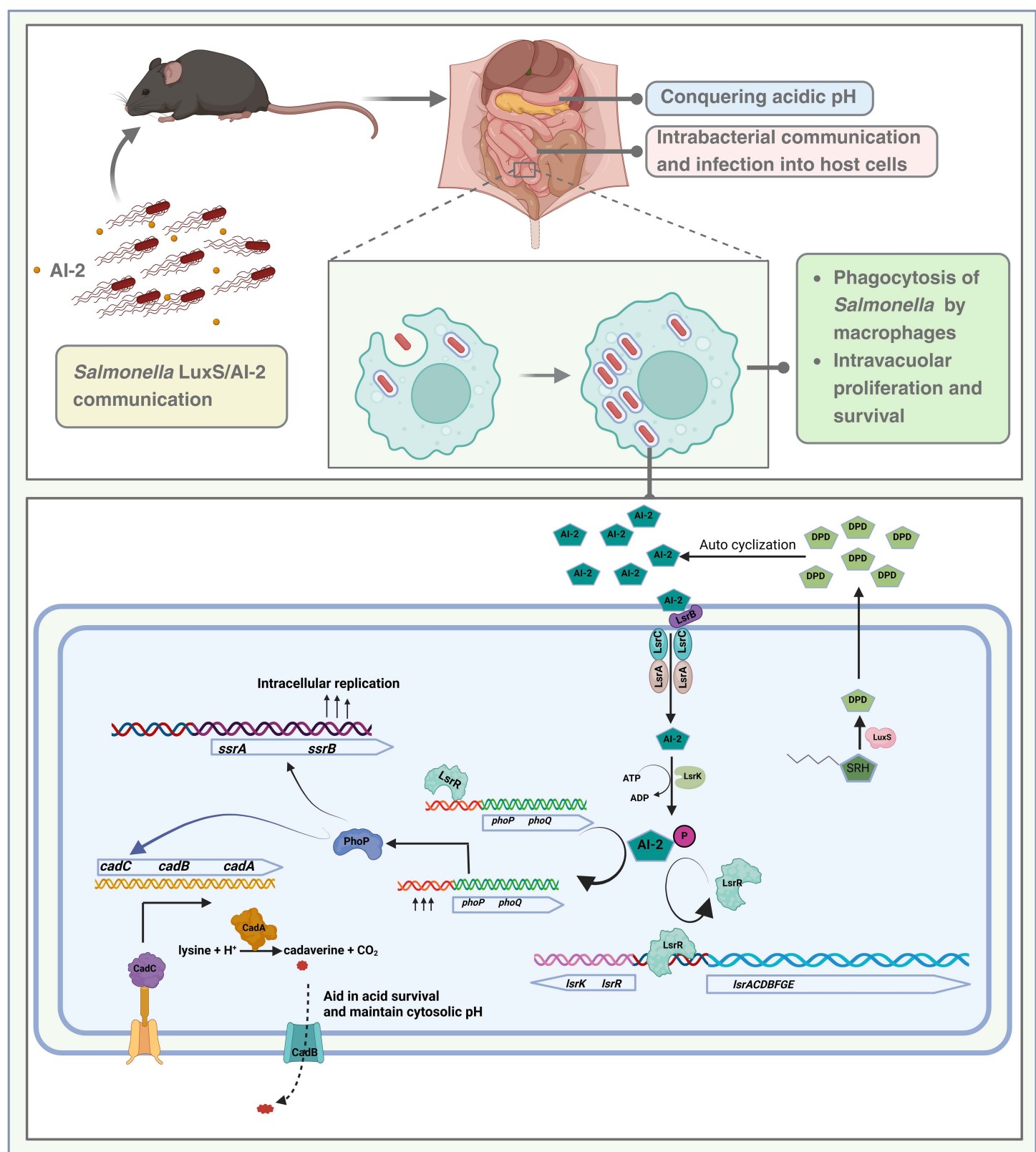

**Fig 7. The proposed model of LuxS/AI-2 signaling to enhance acid stress survival in *Salmonella* Typhimurium.** Created in BioRender. Chakravortty, D. (2026) https://BioRender.com/w2ra69x [46]. LuxS/AI-2 regulates the PhoP/PhoQ two-component system to survive in acidic conditions,

helping in acid adaptation and maintaining cytosolic pH by modulating Cadaverine/Lysine. LsrR, the repressor of the *lsr* operon, also represses the *phoP/phoQ* operon by binding through its Y25 and R43 amino acids. It tunes the expression of SPI-2 genes, thereby enhancing the intracellular division of *Salmonella* in macrophages and *in vivo* colonization.

A growth kinetics study was conducted in LB and M9 minimal medium [48,49] (S2 Table). Briefly, STM WT and mutants were inoculated in fresh LB broth (1:100 ratio from overnight primary culture). The cultures were incubated at 37 ˚C (170 rpm) in an orbital shaker incubator. The OD at 600nm was measured at regular intervals until 24 hours post-inoculation.

## Bacterial gene knockout and strain generation

*Salmonella enterica* serovar Typhimurium strain 14028S and isogenic mutants were used for all assays. *luxS*, *lsrB*, *lsrK,* and *lsrR* gene knockout strains were made by λ-Red recombination system as described previously in one gene step chromosomal gene inactivation method demonstrated by Datsenko and Wanner [50]. Briefly, STM WT strains were transformed with pKD46 plasmid expressing λ-red recombinase system under Arabinose inducible plasmid. The pKD3 and pKD4 plasmid were used as the template to amplify the chloramphenicol and kanamycin resistance cassette with knockout primers (S3 Table). The amplified reaction products were purified using chloroform isopropanol precipitation. The purified PCR product was transformed into pKD46- containing STM WT expressing λ-Red recombinase system. The transformed cells were plated on LB agar with chloramphenicol and kanamycin accordingly. The knockout strains were confirmed by PCR using primers (S3 Table) corresponding to the regions ~100 bp upstream and downstream of the relevant genes.

For complement strain, colony PCR amplified the *luxS* gene with gene-specific primers (containing BamH1 and HindIII restriction site sequence) (S3 Table). The amplified PCR product was purified by chloroform-isopropanol precipitation and cloned into the empty pQE60 vector using BamH1 and HindIII sites. The double-digested insert and vector were subjected to ligation by T4DNA ligase in ligation buffer (NEB) overnight at 16 °C. The respective ligated vector was transformed into the STM Δ*luxS* strains to generate the complemented strain. Complement strain was initially confirmed by colony PCR with cloning primers and internal primers of the genes, and finally by restriction digestion of recombinant plasmid isolated from complement strains to study insert release.

## Cell culture and maintenance

Murine macrophage cell line RAW 264.7 was maintained in Dulbecco's Modified Eagle Medium (DMEM, Lonza), supplemented with 10% FBS (Gibco) and 1% penicillin-streptomycin (Sigma-Aldrich) in a humidified incubator with 5% $CO_2$ at 37 °C. The cells were seeded into the respective cell culture plate for each experiment to conduct intracellular survival assays and intracellular gene expression experiments.

## Peritoneal macrophage isolation

The primary macrophage isolation was performed as previously described [36]. Briefly, C57BL/6J mice (5–6 weeks old) were injected with Brewer's thio-glycolate medium (8% w/v, HIMEDIA). At four days of treatment, 5 ml of cold PBS was injected into the peritoneal cavity, followed by aspiration of the fluid from the cavity and dispensed into a 15 ml centrifuge tube. The centrifugation was done at 300g for 10 min, and the resuspended cell pellet [in Roswell Park Memorial Institute 1640 (RPMI 1640)] was supplemented with 10% FBS and an antibiotic cocktail (penicillin-streptomycin). Cells were counted by hemocytometer and seeded onto 24 well cell culture plates.

## Chemical synthesis of acetonide-protected (*S*)-4,5-dihydroxy-2,3-petanedione (DPD)

Acetonide-protected DPD **6** was synthesized from commercially available *L*-serine **1** in five steps (for details, see the supplemetary Information). These steps include stereoretentive displacement of the amino group of *L*-serine through diazotisation and esterification to form methyl (*S*)-2,3-dihydroxypropanoate **2** [51] followed by protection of 1,2-diol as acetonide (**3** [52]. Transformation of methyl ester of **3** to *N,N*-dimethyl amide **4** [53] paved the way for its conversion to isopropenyl ketone **5** by reacting with isopropenyl Grignard reagent. Ozonolytic cleavage of the alkene in **5** resulted in acetonide-protected (*S*)-DPD **6** [54].

## AI-2 bioassay

The culture of *Salmonella* Typhimurium WT and mutants was obtained at 0h, 1h, 3h, 6h, 9h, and 12h, and centrifuged at 6000rpm for 10min. After filtration using a 0.22 µm filter, the supernatant was stored at -20 °C until required. The AI-2 was quantified using the sensor strain *Vibrio campbellii* ATTC BAA-1117 (strain designation *V. harveyi* BB-170, which only responds to AI-2). The AI-2 bioassay was performed as previously described [1,55] with minor modifications. A single colony of *V. harveyi* BB-170 was incubated in 5ml of Autoinducer Bioassay medium (ATCC Medium: 2746 AB Medium) at 30°C and 200rpm. A 1:1000 dilution was obtained by transferring a 10 µl sensor strain culture to 10mL of fresh AB medium. Then, 20 µl of *Salmonella* cell-free culture supernatant was added to 180 µl diluted culture of the sensor strain. The mixture was added to a white, flat-bottomed 96-well microtiter plate (Corning, USA), followed by shaking in a rotary shaker at 200rpm, at 30 °C. Luminescence was measured at 3h in a fluorescent microtiter plate reader. Autoinducer signal calculated as – (luminescence produced by *Vibrio* in the presence of *Salmonella* Typhimurium free supernatant - luminescence in control medium)/luminescence produced by *Vibrio* with sterile medium. To validate the activity of exogenous DPD molecule and BF-8 inhibitor, an autoinducer bioassay was performed in the presence of DPD and BF-8 inhibitor (Cat# 53796, Sigma). For the intracellular AI-2 assay, RAW 264.7 macrophages were infected, and cell lysates were collected at different time points.

## Gentamicin protection assay

RAW 264.7 macrophages and peritoneal macrophages isolated from C57BL/6J mice were seeded in tissue culture plates for infection. Cells were infected with STM WT, STM Δ*luxS,* STM Δ*lsrB,* STM Δ*lsrK,* STM Δ*lsrR,* and STM Δ*luxS:luxS* (stationary phase culture growing overnight in LB broth). The Multiplicity of Infection (MOI) of 10 was used for the intracellular survival assay. MOI of 20 was used for the *Salmonella* Typhimurium gene expression study (qRT-PCR). Following the infection of STM strains into the cell line, the plate was spun for 5min at 600rpm using a Rota-Superspin R–NV swing

bucket centrifuge. The plate was then incubated for 25 min at 37 °C with 5% $CO_2$ in a humidified incubator. The infected cells were washed twice with 1X PBS and treated with a 100 μg/mL concentration of gentamycin. After 1h, the medium were changed with a reduced concentration of gentamycin (25 μg/mL) and incubated until the desired time point. The time points of 2h, 6h, and 16h were taken for the qRT-PCR samples and 2h and 16h for the intracellular survival assay.

Also, for studying the effect of exogenous DPD in the recovery of the phenotype of mutant *luxS,* were grown *in vitro* with supplementation of 25 μM, 50 μM,100 μM, and 200 μM DPD molecules and 10%, 20%, 30%, and 50% spent medium of STM WT collected at 3h in LB growing. For inhibition, the inhibitor (Z)-4- bromo-5-(bromomethylene)-3-methylfuran-2(5H), known as BF-8 treatment, was added to STM WT in LB medium. DPD or spent medium supplemented and BF-8 treated culture was used for gentamycin protection assay.

### Intracellular survival assay and phagocytosis assay

The cells were lysed with 0.1% triton-X 100 in PBS at specific time points post-infection. For the intracellular survival assay, 2h and 16h post-infection samples were collected and plated, and the corresponding CFU at 2h and 16h was determined. Fold proliferation was determined using the formula [CFU at 16 h]/[CFU at 2 h]. For phagocytosis, the bacterial number was determined in the inoculum and 2h time point post-infection. Percent phagocytosis was by using the formula- (CFU at 2 h)/(CFU of pre-inoculum)]×100.

### Confocal microscopy

To study the *Salmonella* proliferation and their intracellular vacuolar status, all STM strains (STM WT, STM Δ*luxS,* STM Δ*lsrB,* STM Δ*lsrK,* and STM Δ*lsrR*) were transformed with pFPV 25.1-mCherry-Amp$^R$ plasmid, and used to infect RAW264.7 cells. Briefly, $1 \times 10^5$ RAW 264.7 cells were seeded on the coverslip and infected with different strains of bacteria at MOI 25. After appropriate post-infection durations, cells were washed twice with PBS and fixed with 3.5% paraformaldehyde (PFA), and incubated with a specific antibody (LAMP1 monoclonal antibody-clone 1D4B; sc-19992, Santa Cruz Biotechnology, USA) in a blocking buffer containing 2% BSA and 0.01% saponin for 2 hours at RT or overnight at 4°C. The cells were washed twice with PBS and incubated with the appropriate secondary antibody conjugated to a fluorochrome (Alexa Fluor 488 or DyLight 488, dilution 1:200) for 1 hour at RT. The coverslips were then mounted onto a clean glass slide and imaged under a confocal microscope (Zeiss LSM-710) using a 63X objective.

### *In vitro* Cell culture competition assay

RAW 264.7 macrophages were infected at a 1:1 ratio with STM WT and STM Δ*luxS.* At 16h, post-infection cells were lysed, and the sample was collected as described above. The competition was also done with the coculture of STM WT and STM Δ*luxS*. This involved the co-cultivation of STM WT and STM Δ*luxS* together, followed by infection in RAW 264.7 macrophages. The competitive index was calculated for a WT strain and Δ*luxS* by dividing the ratio between CFU (STM Δ*luxS*) and CFU (STM WT) by the ratio of both strains in the inoculum.

### RNA isolation and RT-qPCR

For gene expression study in Luria Bertani (LB) medium and F-medium (acidic media that mimics the cell vacuole environment) [49,56]. An overnight-grown culture in LB medium was subjected to subculture (at 1:100) in LB or F-medium. At 3h, 6h,9h, and 12h, bacterial cell pellets were resuspended in TRIzol (from TaKaRa, RNA isoPlus- 9109) and kept at -80˚C. Total RNA was isolated by chloroform extraction followed by isopropanol precipitation. To evaluate the quality, the amount of RNA was quantified in nanodrop (ThermoFischer) and examined on a 2% agarose gel.

To produce cDNA, 3 μg of RNA sample was treated with DNase I (TaKaRa) at 37˚C for 1h followed by heating for 10 min at 65˚C. PCR using 16S primers showed no amplification in DNase-treated samples, confirming the absence of

genomic DNA contamination and validating RNA-specific quantification. DNA-free RNA samples were used to make cDNA using the PrimeScript RT reagent Kit provided by TaKaRa (Cat# RR037A).

For gene expression in *Salmonella* Typhimurium upon infection in RAW 264.7 macrophages, infection was performed at MOI of 20. At 2h, 6h, and 16h post-infection, the cells were lysed using TRIzol and kept at -80˚C. Total RNA isolation and cDNA synthesis were carried out using the manufacturer's protocol. The list of expression primers is provided in S3 Table. Relative gene expression was determined using the ΔΔCt method, with the 3h time point (LB culture) and 2h post-infection (macrophage assays) used as the respective reference conditions for normalization. For comparative analyses, expression levels were calculated relative to STM WT, which served as the reference strain.

## Acid survival assay

Overnight cultures of STM WT, STM Δ*luxS,* STM Δ*lsrB,* STM Δ*lsrK,* and STM Δ*lsrR* were adjusted to $OD_{600}$ - 0.3. For survival in 1XPBS, pH was adjusted to 3,4,5,6,7, or 8 using concentrated HCl. Strains of *Salmonella* Typhimurium ($10^7$ bacteria/ml) were exposed to different pH as mentioned in 1X PBS and incubated for 2h. At 2h post-treatment, CFU were enumerated by plating on LB agar medium. The same protocol was followed for survival in LB medium with similar pH ranges. For the strain of STM harboring plasmids pQE60 complemented with *phoP/phoQ*, a survival assay was done in pH of 3, 4, and 5 in 1X PBS.

## Acid tolerance assay

A standard acid tolerance assay was performed as previously described with slight modification [26]. Briefly, acid tolerance assay (ATR) was conducted with strains grown overnight at 37°C in LB broth containing the appropriate antibiotic. The culture corresponding to 0.3 $OD_{600}$ of the overnight culture (unadapted) was centrifuged and the pellet was resuspended into 2ml of LB broth (pH 5) and incubated at 37°C with shaking for 2h (known as adapted culture). The acid challenge of unadapted and adapted cultures involved readjusting the pH to 3. CFU was determined for adapted and unadapted cultures after exposure to pH 3 (2h of pH 3 treatment). ATR assay was also done with the strain of *Salmonella* containing plasmid pQE60 complemented with *phoP/phoQ* gene*.*

## Intracellular pH measurement by pHuji plasmid

The plasmid pBAD:pHuji (Plasmid #61555) was used to transform STM WT, STM Δ*luxS*, STM Δ*lsrB*, STM Δ*lsrK*, and STM Δ*lsrR*. pHuji is a pH-sensitive red fluorescent protein demonstrating a more than 20-fold fluorescent intensity change from pH 5.5 to 7.5 [57]. Overnight-grown bacterial strains were subjected to a pH range of 3.0 to 8.0 in phosphate buffer (PBS) in 40 µM of sodium benzoate to ensure equilibration to the desired pH. The resulting fluorescence intensity ratio, obtained from flow cytometric analysis, was plotted as a function of pH and fitted to get a standard curve. The standard curve was used to interpolate ratios measured within the intracellular pH of bacteria in LB medium, F-medium, and RAW 264.7 macrophages.

## Mass spectrometry for the determination of lysine and cadaverine

Overnight grown STM WT, STM Δ*luxS*, STM Δ*lsrK,* STM Δ*lsrB,* and STM Δ*lsrR* were sub-cultured in acidic pH 5 LB medium and incubated at 37°C until mid-log phase. Thereafter, bacteria were spun down and the spent medium were collected. Spent medium were kept with an equal volume of acetone at -20 °C for overnight. Next, samples were collected after centrifugation at 13k rpm for 10min. Samples were analysed in Orbitrap fusion connected to UHPLC Vanquish (Thermo Scientific). Water and acetonitrile with 0.1% formic acid were used as a mobile phase. The flow rate was kept at 0.3ml/min linear gradient was started with 5% to 95% using Hypersil Gold C18 column (2.1×100mm & particle size 1.9µ).

## Plasmid construction, and site-directed mutagenesis

The vector pET28a(+) was used to construct plasmids p*lsrR* for the overexpression of LsrR as a C-terminal 6xHis-tagged fusion protein. The DNA fragment containing *the lsrR* coding region was amplified by polymerase chain reaction (PCR) from the chromosome of *Salmonella* Typhimurium 14028s using primers (F-lsrR and R-lsrR containing restriction site NcoI and XhoI, respectively). The PCR products were digested with NcoI and XhoI, and ligated with NcoI and XhoI digested pET28a(+). The inserts were checked by DNA sequencing.

For Y25A and R43A site-directed mutagenesis in LsrR protein, PCR for SDM was set up with template DNA from pETa(+)- *lsrR* and amplified with Q5 polymerase (NEB). The PCR products were digested with DpnI and transformed into *E. coli* DH5-alpha. Y25A and R43A double mutants were generated using pET28a(+) *lsrR* R43A plasmid as a template for SDM using primers specific to the second set of mutations. All plasmids were confirmed by DNA sequencing.

## Overexpression and purification of LsrR

LsrR protein was purified by using the protocol as described previously [58]. pET28a(+)-*lsrR* and its mutants containing C-terminal 6xHis tag were transformed into *E. coli* Rosetta (DE3). The transformed colonies were inoculated into 5 ml LB containing kanamycin (Kan) and chloramphenicol (Cm) and grown overnight. Inoculum (1%) was added to 1.2 L LB containing Kan and Cm, grown at 37 °C to an $OD_{600}$ of 0.4 under shaking, and thereafter supplemented with 0.1 mM IPTG, and grown at 16°C overnight. Cells were harvested by centrifugation at 4°C, resuspended in 10 ml buffer A [20 mM Tris–HCl pH 8, M NaCl, 10% glycerol (v/v), 1.5 mM β-mercaptoethanol, 1 mM PMSF] lysed by sonication, and centrifuged at 4°C in a pre-cooled centrifuge at 10000 rpm for 10 min. The supernatant was loaded onto a 1 ml Ni-NTA column equilibrated with buffer A, washed with 20 ml buffer A, and eluted with a gradient of imidazole (20–500 mM) in the same buffer. The fractions were analysed on 12% SDS-PAGE. Fractions enriched for LsrR were pooled, loaded onto a Superdex75 gel filtration column, and eluted in buffer B [20 mM Tris–HCl pH 8, 1M NaCl, 10% glycerol (v/v) and 1.5 mM β-mercaptoethanol]. The purity of LsrR was checked on 12% SDS-PAGE (S7H Fig). Fractions with apparent homogeneity were pooled, concentrated using a 10 kDa cut-off Centricon (Millipore), and estimated by Bradford's method using bovine serum albumin (BSA) as standard. The proteins were dialyzed against buffer A containing 50% glycerol (v/v) and stored at –20° C.

## Gel shift assay

HPLC-purified 90 bp single-strand DNA oligos of the *lsrR* and *phoP* operon promoters (Sigma) were used for a gel shift assay. Binding reactions involved incubating promoter DNA with varying concentrations of LsrR or its mutated form. The incubation buffer contained 50 mM Tris-Cl (pH 7.5), 150 mM NaCl, 3 mM magnesium acetate, 0.1 mM EDTA, and 0.1 mM DTT. After 15 min at room temperature, the mixture was combined with gel loading buffer (60% 0.25×TBE, 40% glycerol, 0.2% bromophenol blue) and run on a 6% native polyacrylamide gel. The staining was done for DNA with SYBR Gold Nucleic Acid Gel Stain (Invitrogen) and analyzed as per the manufacturer's instructions.

## *In vivo* animal experiment

Oral gavaging of $10^7$ CFU of STM WT, STM Δ*luxS*, STM Δ*lsrB*, STM Δ*lsrK*, STM Δ*lsrR*, and STM Δ*luxS:luxS* was used to infect 5- to 6-week-old C57BL/6J mice. Five days post-infection, the intestine (Peyer's patches), MLN (mesenteric lymph node), spleen, liver, and blood were aseptically extracted (in a Biosafety level 2 cabinet) to examine the colonization in the organs. The CFU were enumerated on differential and selective *Salmonella-Shigella* (SS) agar.

## Mice survival assay

Male C57BL/6, aged 5–6 weeks, were obtained from the Central Animal Facility, IISc. The mice were given $10^8$ CFU of each strain (overnight grown cultures) orally to compare the survival and weight alterations of mice post-infection.

Following infection, the mice were observed every day to determine their survival and weight, and the results were expressed as a percentage of survival (Kaplan-meier curve) and weight reduction.

All experiments comply with the rules set forth by the Indian Institute of Science, Bangalore's IAEC. The approved protocol number is CAF/Ethics/852/2021. The Institutional Animal Ethics Committee approved every animal experiment, and the National Animal Care Guidelines were scrupulously followed.

## Statistical analysis

As mentioned in the figure legends, each experiment has been independently repeated two to five times. GraphPad Prism 8.4.3 was utilized for all statistical analyses. Normality of the data was assessed using the Shapiro–Wilk test. In addition, Q–Q plots were visually inspected to evaluate deviations from normality, particularly given the small sample size. Parametric tests were applied only when data did not significantly deviate from normality ($p > 0.05$). As the figure legends state, the statistical analyses included an unpaired, two-tailed Student's t-test, One-way ANOVA with Dunnett's post hoc test, and Two-way ANOVA with Tukey's post hoc test. A non-parametric one-way ANOVA (Kruskal-Wallis) test with Dunn's post-hoc test was performed for the animal experiment. P-values less than 0.05 were regarded as significant. The analysis is presented as mean ±SD with information on group sizes and p values mentioned in the respective figure legends.

## Supporting information

**S1 Fig. LuxS/AI-2 signaling is not required for *in vitro* growth but is essential for intracellular survival. (A)** *Salmonella* growth with AI-2 production in LB medium, **(B)** STM WT, STM *ΔluxS,* and STM *ΔlsrB* AI-2 production in LB medium (data is from one experiment representative of 3 independent experiments). **(C)** The mRNA expression of genes *lsrB*, *lsrK*, and *lsrR,* and **(D)** *luxS* gene expression in STM WT *in vitro* growth in LB (normalization relative to 3h) (data is from one experiment representative of 3 independent experiments). Growth kinetics study upon deletion of gene *luxS, lsrB, lsrk,* and *lsrR* **(E)** LB medium, **(F)** Minimal medium (data is from one experiment representative of 3 independent experiments). **(G)** 2h post-infection and **(F)** 16h post-infection, Fluorescence microscopy of STM infection in RAW 264.7 macrophages. Representative of N = 2, n ≥ 10.
(TIF)

**S2 Fig. Spent medium or exogenous AI-2 enhances STM survival in macrophages, while inhibition of AI-2 signaling attenuates it. (A)** Percentage phagocytosis **(B)** Fold proliferation in RAW 264.7 macrophages of STM *ΔluxS* upon treatment of STM WT spent medium at 10%, 20%, and 30%. **(C)** Percentage phagocytosis, **(D)** Fold proliferation of STM in RAW 264.7 macrophages upon treatment of synthetic DPD molecule. **(E)** Survival assay of STM WT and STM *ΔluxS* in the presence of inhibitor BF-8. **(F)** Autoinducer assay- Light production by *Vibrio* BB170 for synthetic DPD molecule. **(G)** Luminescence by *Vibrio* BB170 in STM WT spent medium (3h growth spent medium) with or without inhibitor BF-8. **(H)** Percentage phagocytosis, **(I)** Fold proliferation of STM in peritoneal macrophages upon treatment of DPD and BF-8 inhibitor. One-way ANOVA with Dunnet's post-hoc test was used to analyze the data; p values **** $p < 0.0001$, *** $p < 0.001$, ** $p < 0.01$, * $p < 0.05$. (All data is from one experiment representative of independent experiments, N ≥ 2, n ≥ 3).
(TIF)

**S3 Fig. AI-2 signaling is induced in an acidic environment and increases survival at acidic pH. (A)** *luxS* **(B)** Gene *lsrB*, *lsrK* and *lsrR* expression study in STM WT upon growth in F-medium (pH 5). STM WT, STM *ΔluxS*, STM *ΔlsrB*, STM *ΔlsrK,* and STM *ΔlsrR* survival in **(C)** PBS and (D) LB medium with a different range of pH [3–8]. **(E)** Survival at pH3 LB medium **(F)** Survival at pH 4 LB media **(G)** Survival at pH5 LB medium. One-way ANOVA with Dunnett's post-hoc test was used to analyze the data; Two-way ANOVA was used to analyze the grouped data; p values **** $p < 0.0001$, *** $p < 0.001$, ** $p < 0.01$, * $p < 0.05$. (Data is from one experiment representative of 3 independent experiments, N = 3, n ≥ 2).
(TIF)

**S4 Fig.** *phoP* **expression is regulated by LuxS/AI-2 signaling and helps in maintaining the cytosolic pH.** mRNA expression of *phoP* gene in STM WT **(A)** LB medium, (normalization relative to 3h) **(B)**LB media (acidic pH5), (normalization relative to 3h) **(C)** F-medium (pH5), **(D)** mRNA expression of *phoP* gene upon infection in RAW 264.7 macrophages (normalization relative to 2h). All data in **A-D** is from one experiment representative of 3 independent experiments). **(E)** mRNA expression of *phoP* in STM WT, STM Δ*luxS*, STM Δ*lsrB*, and STM Δ*lsrK* (normalization relative to 3h). **(F)** *phoP* gene expression in STM WT, STM Δ*luxS*, STM Δ*lsrB*, and STM Δ*lsrK*, in F-medium (pH 5). **(G)** mRNA expression of *phoP* in STM Δ*luxS* upon treatment of STM WT spent medium (neutral). Standard curve of amcyan/FITC ratio in the range of pH 3–8 PBS of phuji plasmid containing STM strains **(H)** STM WT, **(I)** STM Δ*luxS*, **(J)** STM Δ*lsrB*, **(K)** STM Δ*lsrK, and* **(L)** STM Δ*lsrR.* One-way ANOVA with Dunnett's post-hoc test was used to analyze the data; Two-way ANOVA was used to analyze the grouped data; p values **** $p < 0.0001$, *** $p < 0.001$, ** $p < 0.01$, * $p < 0.05$. All data are represented as mean ± SD from independent experiments (N = 3, n ≥ 3).
(TIF)

**S5 Fig. PhoP regulates** *cadBC/A* **genes to maintain cytosolic pH.** mRNA expression of **(A)** *cadA,* **(B)** *cadB,* and **(C)** *cadC* gene in STM WT and STM Δ*phoP upon* infection in RAW 264.7 macrophages. Two-way Anova was used to analyze the grouped data; p values **** $p < 0.0001$, *** $p < 0.001$, ** $p < 0.01$, * $p < 0.05$. All data are represented as mean ± SD from independent experiments (N = 3, n = 3).
(TIF)

**S6 Fig. LsrR controls** *phoP* **expression by binding to its promoter. (A,B)** Superimposed close-up structural models showcasing key interactions between LsrR (green) residues and the *phoP* and *lsr* promoters (yellow), emphasizing critical contact points. Three individual AlphaFold3 structural models with the highest prediction rankings were aligned, and the interacting residues are displayed in a ball-and-stick representation. **(C, D)** Quantification of representative gel image of Fig 4C and 4D, and extended Fig 6E and 6F. Quantification done as the area under the curve (AUC) of protein-DNA bound/(AUC protein DNA bound +AUC protein Unbound DNA). Electrophoretic Mobility Shift Assay (EMSA) Y25A LsrR and R43A LsrR protein with double-stranded **(E)** *lsr* promoter and **(F)** *phoP* promoter.
(TIF)

**S7 Fig. LsrR interacts with the** *lsr* **and** *phoP* **promoters through the Y25 and R43 amino acid residues.** Electrophoretic Mobility Shift Assay (EMSA) of *lsr* promoter with **(A)** R43A LsrR and **(B)** Y25A/R43A LsrR protein in increasing concentrations of NaCl in the binding buffer. EMSA of WT LsrR, Y25A/R43 LsrR, Y25A LsrR, and R43A LsrR with single-stranded (90 bp) *lsr* promoter **(C,E)** and *phoP* promoter**(D,F)**. EMSA of WT LsrR protein with random 60 bp DNA sequence **(G)**. SDS-PAGE of purified protein (1ug protein of each calculated by Bradford assay, molecular weight of LsrR ~ 35kDa) **(H)**.
(TIF)

**S8 Fig.** *phoP* **complementation under a non-native promoter in STM Δ***luxS* **rescued impaired survival in macrophages. (A)** Percentage phagocytosis, **(B)** Fold proliferation, of STM WT, STM WT: pQE60-*phoP/phoQ,* STM Δ*luxS,* STM Δ*luxS:* pQE60-*phoP/phoQ* upon infection in RAW 264.7. One-way ANOVA with Dunnett's post-hoc test was used to analyze the data. (Data is from one experiment, representative of 3 independent experiments).
(TIF)

**S9 Fig. LuxS/AI-2 signaling regulates** *ssrB/ssrA* **gene expression through PhoP.** mRNA expression of **(A)** *ssrA,* **(B)** *ssrB* gene in STM WT and STM Δ*phoP upon* infection in RAW 264.7 macrophages. **(C)***Tnf-α,* **(B)** *Il-10* gene expression in RAW 264.7 macrophages upon infection of STM WT STM Δ*luxS,* STM Δ*lsrB,* STM Δ*lsrK.* **(E) and (F)** mRNA expression of *phoP* gene in STM WT: pQE60-*phoP/phoQ,* Δ*luxS*: pQE60-*phoP/phoQ* upon infection in RAW 264.7 macrophages. *ssrA* gene expression in *phoP/phoQ* cloned strain **(E)** STM Δ*luxS,* **(F)** STM Δ*lsrB,* **(G)** STM Δ*lsrK*. One-way ANOVA with

Dunnett's post-hoc test was used to analyze the data. Two-way Anova was used to analyze the grouped data; p values **** $p < 0.0001$, *** $p < 0.001$, ** $p < 0.01$, * $p < 0.05$. All data are represented as mean ± SD from independent experiments (N = 3, n = 3).
(TIF)

**S10 Fig. LuxS/AI-2 signaling facilitates colonization at secondary organ sites. (A)** The experimental protocol for organ burden in C57BL/6J mice by peritoneal infection of $10^4$ CFU per mouse. The organ burden post 3 days of peritoneal infection **(B)** Spleen, **(C)** Liver, and dissemination in **(D)** Blood. Represented as Mean +/-SD of N = 2, n = 5. **(E)** C57BL/6J mice were infected by orally gavaging $10^8$ CFU per mouse. Mice weight (in grams) was noted on each day of post-infection. **(F)** C57BL/6J mice infected by orally gavaging $10^8$ CFU per mouse and BF-8 inhibitor treatment were given on an alternate day of infection. Mice weight was noted on each day of post-infection. Represented as Mean +/-SD of N = 2, n = 5. **(G)** The hematoxylin and eosin staining of the sections of the liver of C57Bl/6 mice infected by orally gavaging $10^7$ CFU per mouse and the BF-8 inhibitor. treatment was given on the alternative day of infection. One-way ANOVA (Kruskal Wallis) with Dunn's post-hoc test was used to analyze organ burden in mice.
(TIF)

**S11 Fig. *phoP* complementation under a non-native promoter in STM Δ*luxS* increased its reduced colonization.** The organ burden post 5 days of oral gavage in **(A)** Intestine, **(B)** Mesenteric lymph node (MLN), **(C)** Spleen, **(D)** Liver, and dissemination in **(E)** Blood. Represented as Mean±SD of n = 5 **(F)** Percentage survival and **(G)** Mice's weight (in grams) upon infection with STM WT, STM WT: pQE60-*phoP*/*phoQ,* STM Δ*luxS,* STM Δ*luxS:* pQE60-*phoP*/*phoQ.* Non-parametric One-way ANOVA (Kruskal-Wallis) with Dunn's post hoc test was used to analyze organ burden in mice.
(TIF)

**S1 Text. Chemical synthesis of acetonide-protected (*S*)-4,5-dihydroxy-2,3-petanedione (DPD).**
(DOCX)

**S1 Table. All the Bacterial strains and plasmids used in this study are given in Table.**
(DOCX)

**S2 Table. Media Composition.**
(DOCX)

**S3 Table. Primers used in this study.**
(DOCX)

## Acknowledgments

Dr. Shashank Tripathi (CIDR, IISc Bangalore) and Dr. Sandeep M Eswarappa (Dept. of Biochemistry, IISc Bangalore) are duly acknowledged for the Luminometer. Divisional Mass Spectrometry facility, IISc, and Mrs. Sunita Joshi for the MS analysis. The Departmental Confocal Facility, Departmental Real-Time PCR Facility, Divisional Flowcytometry Facility, and Central Animal Facility at IISc are duly acknowledged. Mr Sumith and Ms Navya are acknowledged for their help in image acquisition. Dr Ritika Chatterjee and Ms Sagrika are acknowledged for their technical help. Schematic images were created with BioRender and can be accessed from Chakravortty, D. (2026) https://BioRender.com/w2ra69x.

## Author contributions

**Conceptualization:** Anmol Singh, Dipshikha Chakravortty.

**Formal analysis:** Anmol Singh, Dipshikha Chakravortty.

**Funding acquisition:** Dipshikha Chakravortty.

**Investigation:** Anmol Singh, Abhilash Vijay Nair, Shashanka Aroli, Raju S. Rajmani, Santanu Mukherjee, Dipshikha Chakravortty.

**Methodology:** Anmol Singh, Abhilash Vijay Nair, Shashanka Aroli, Suman Das, Subhrajit Karmakar, Raju S. Rajmani, Santanu Mukherjee, Umesh Varshney, Dipshikha Chakravortty.

**Project administration:** Anmol Singh, Dipshikha Chakravortty.

**Resources:** Santanu Mukherjee, Umesh Varshney, Dipshikha Chakravortty.

**Supervision:** Santanu Mukherjee, Umesh Varshney, Dipshikha Chakravortty.

**Validation:** Anmol Singh, Abhilash Vijay Nair, Shashanka Aroli, Suman Das, Subhrajit Karmakar, Raju S. Rajmani, Santanu Mukherjee, Umesh Varshney, Dipshikha Chakravortty.

**Visualization:** Anmol Singh, Abhilash Vijay Nair, Shashanka Aroli, Santanu Mukherjee, Umesh Varshney, Dipshikha Chakravortty.

**Writing – original draft:** Anmol Singh, Shashanka Aroli, Santanu Mukherjee, Dipshikha Chakravortty.

**Writing – review & editing:** Anmol Singh, Abhilash Vijay Nair, Shashanka Aroli, Santanu Mukherjee, Umesh Varshney, Dipshikha Chakravortty.

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
