## [Decision Letter · Decision Letter 0]

16 Dec 2025

PPATHOGENS-D-25-02450

The quorum-sensing lexicon of Salmonella ameliorates acid stress in the host by a non-canonical mechanism

PLOS Pathogens

Dear Dr. Chakravortty,

Thank you for submitting your manuscript to PLOS Pathogens. After careful consideration, we feel that it has merit but does not fully meet PLOS Pathogens's publication criteria as it currently stands. Therefore, we invite you to submit a revised version of the manuscript that addresses the points raised during the review process.

We look forward to receiving your revised manuscript.

Kind regards,

Camila Valenzuela

Guest Editor

PLOS Pathogens

Matthew Wolfgang

Section Editor

PLOS Pathogens

Sumita Bhaduri-McIntosh

Editor-in-Chief

PLOS Pathogens

orcid.org/0000-0003-2946-9497

Michael Malim

Editor-in-Chief

PLOS Pathogens

orcid.org/0000-0002-7699-2064

**Additional Editor Comments :**

Dear Dr. Chakravortty,

Thank you very much for submitting your manuscript "The quorum-sensing lexicon of Salmonella ameliorates acid stress in the host by a non-canonical mechanism" (PPATHOGENS-D-25-02450) for review by PLOS Pathogens. Your manuscript was fully evaluated at the editorial level and by three independent peer reviewers. The reviewers appreciated the attention to an important problem, but raised some substantial concerns about the manuscript as it currently stands. These issues must be addressed before we would be willing to consider a revised version of your study. We cannot, of course, promise publication at this time. We therefore ask you to modify the manuscript according to the review recommendations before we can consider your manuscript for acceptance. Your revisions should address the specific points made by each reviewer.

I am returning your manuscript with three reviews. The reviewers all agree that this work is new and interesting information for the field, but requires clarifications on the rationale and the way some some experiments were performed, analyzed and presented, and substantial language editing is needed to improve clarity and readability. After reading the reviews and looking at the manuscript, I recommend Major Revision based on the critiques from the more critical reviews. I am sorry I cannot be more positive at the moment, however we are looking forward to receiving your revision. With a a lot of work, the manuscript will be suitable for a resubmission, if you so wish to do so.

Note that we will send your paper back to the same reviewers upon resubmission. Please pay particular attention to the following reviewer suggestions and give them due consideration:

Reviewer 1 - some experiments should be reproduced in media different to the presented. Data visualization and interpretation should be revised to ensure accurate conclusions are drawn.

Reviewer 3 - please revise the statistics used for the analysis throughout the manuscript. Some of the expertiments should be repeated to include technical controls. Some of the data should either be revised or re-analyzed.

(1) A letter containing a detailed list of your responses to the review comments and a description of the changes you have made in the manuscript.

(2) Two versions of the manuscript: one with either highlights or tracked changes denoting where the text has been changed; the other a clean version (uploaded as the manuscript file).

We hope to receive your revised manuscript within 60 days. If you anticipate any delay in its return, we ask that you let us know the expected resubmission date by replying to this email. Revised manuscripts received beyond 60 days may require evaluation and peer review similar to that applied to newly

submitted manuscripts.

We are looking forward to receiving your revision.

Sincerely,

Camila Valenzuela, PhD

Guest Editor

PLOS Pathogens

Matthew C. Wolfgang, Ph.D.

Section Editor

PLOS Pathogens

**Journal Requirements:**

1) Please provide an Author Summary. This should appear in your manuscript between the Abstract (if applicable) and the Introduction, and should be 150-200 words long. The aim should be to make your findings accessible to a wide audience that includes both scientists and non-scientists. Sample summaries can be found on our website under Submission Guidelines:

https://journals.plos.org/plospathogens/s/submission-guidelines#loc-parts-of-a-submission

- TM on page: 32.

3) We notice that your supplementary Tables are included in the manuscript file. Please remove them and upload them with the file type 'Supporting Information'. Please ensure that each Supporting Information file has a legend listed in the manuscript after the references list.

Potential Copyright Issues:

i) Figures 6A, 6G, 6O, and S9A. Please confirm whether you drew the images / clip-art within the figure panels by hand. If you did not draw the images, please provide (a) a link to the source of the images or icons and their license / terms of use; or (b) written permission from the copyright holder to publish the images or icons under our CC BY 4.0 license. Alternatively, you may replace the images with open source alternatives. See these open source resources you may use to replace images / clip-art:

Note: If the figure is created through BioRender. Please state this in the figure legend. Please also confirm that you hold a Premium account and provide a pdf copy of the CC BY 4.0 Licence as provided by BioRender. For instructions on how to generate a CC BY 4.0 license for your figure, please see the guidelines here: https://help.biorender.com/hc/en-gb/articles/21282341238045-Publishing-in-open-access-resources.

If you are using the free assets from BioRender, we are unable to publish these images as they are licenced under a stricter licence than CC BY 4.0. In this case we ask you to remove the BioRender images and replace them with open source alternatives.

See these open source resources you may use to replace images / clip-art:

- https://bioart.niaid.nih.gov/

- https://bioicons.com/

- https://healthicons.org/

- https://scidraw.io/

- https://reactome.org/icon-lib

- https://www.phylopic.org/images

- https://journals.plos.org/plosbiology/article?id=10.1371/journal.pbio.3002395

5) We note that your Data Availability Statement is currently as follows: "The data is in the manuscript.". Please confirm at this time whether or not your submission contains all raw data required to replicate the results of your study. Authors must share the “minimal data set” for their submission. PLOS defines the minimal data set to consist of the data required to replicate all study findings reported in the article, as well as related metadata and methods (https://journals.plos.org/plosone/s/data-availability#loc-minimal-data-set-definition).

7) Please ensure that the funders and grant numbers match between the Financial Disclosure field and the Funding Information tab in your submission form. Note that the funders must be provided in the same order in both places as well.

**Reviewers' Comments:**

Reviewer's Responses to Questions

**Part I - Summary**

Reviewer #1: This work investigates the importance of the Salmonella lsr quorum-sensing system of the ability of this pathogen to survive in macrophages and under acidic conditions. It demonstrates that mutants unable to produce (luxS), transport (lsrB), or phosphorylate (lsrK) the AI-2 signal, but not the repressor of the regulon (lsrR), are defective in these functions. The work implicates the PhoQ/PhoP regulator as the site of action of LsrR, demonstrating binding to the phoP promoter region that is absent in an LsrR mutant with two amino acid changes. In vivo studies using a mouse model of oral or peritoneal infection support the physiological importance of this control. Overall, the work produces new and interesting information, with some questions about techniques used and their interpretation.

Reviewer #2: In their research, the authors examine the non-canonical quorum-sensing mechanism in Salmonella. The paper is clearly written and well organized, systematically explaining the initial thesis and demonstrating the proposed mechanism.

My questions and concerns are listed below:

Introduction:

The introduction section needs some reorganization; too many repetitive sentences make it difficult to read and a bit too long. Overall, it provides the necessary background information, but in a scattered manner.

Results:

The justification and explanation of each part of the results section are too elaborate. Some of it should be placed in either the introduction or the discussion section. This makes the results section too long, and some information is repeated several times throughout the manuscript.

Line 234: “Our study shows that environmental acidification is the critical physiological signal that upregulates AI-2 production by Salmonella (Fig. 2G).”

I am not sure how this figure is showing the above-mentioned observation. Please clarify. Upregulation compared to what exactly?

Figures should be corrected (graphical point of view: figure alignment).

Discussion

In the discussion section, the authors extensively cover microbiome composition; however, their focus is primarily on Salmonella mechanisms of communication and QS, with no direct experiments linked to or correlated with the microbiome study. Therefore, I recommend shortening this section of the discussion.

As a summary, I have to admit that I read this paper with a large dose of satisfaction. It is a well-planned and excellently executed study, which brings new data and knowledge to the field. It should also be of interest to the general public, as it addresses urgent topics in bacterial infection, bacterial communication, and potential antimicrobial strategies.

Reviewer #3: The authors of the paper ‘The Salmonella quorum sensing lexicon attenuates acid stress in the host through a non-canonical mechanism’ demonstrate the role of the AI-2 autoinducer-dependent quorum sensing system in Salmonella's response to acid stress via the two-component PhoP/PhoQ system and subsequent regulation of SPI-2 and acid tolerance response genes. The phenotypic impact on the intracellular survival of Salmonella in macrophages and in vivo in a mouse model is also described. A great deal of work is presented. However, the paper is difficult to read and the logic behind the sequence of results is not always clear, especially at the beginning. That is why I would like to recommend the publication of this work, but only after answering the specific points below and after a fairly substantial rewriting process. I hope that my comments/suggestions will help the authors to improve their article.

**Part II – Major Issues: Key Experiments Required for Acceptance**

Reviewer #1: 1 .Fig. 1A and Fig. 2G show production of AI-2 during the growth cycle. Fig. 2G, however, does not show the growth of the cultures. An appropriate way to show these data and compare between them would be to use normalized luminescence (RLU/OD600). This would allow the reader to assess the differences in AI-2 production per bacterium.

2. Fig. 2A: The suggestion is that the luxS mutant cannot use AI-2 produced by the wild type in a macrophage. The more likely explanation is that the two strains did not reside in the same phagolysosome. This is mentioned in the Discussion but should be the primary conclusion here.

3. Fig. 2G-J shows the effects of pH on AI-2 production and its consequences on bacterial survival using F-medium adjusted to various pH levels. These results, however, are compared to those of Fig. 1, for which LB medium was used. The experiments should all use the same medium with pH adjustments, as the two are very different from each other; either LB at low pH, or F-medium at neutral pH.

4. Fig. 3A needs to be better labeled. It also might be of greater value if it were presented earlier in the manuscript.

5. Fig. 3BC tests expression of phoP in LB medium. Is not phoP poorly expressed under these conditions, requiring low pH, magnesium, and iron for induction? Using phoP-inducing conditions would be appropriate.

6. Fig. 3G attempts to determine the cytoplasmic pH using the pHuji plasmid. These presumed pH values are, in fact, extrapolated from those obtained from culture of the bacteria in media with various pH values. Growth at a specific pH, however, does not equate to cytoplasmic pH, as bacteria possess extensive systems to buffer the cytoplasm. The approach may be helpful in some relative way to demonstrate acidification of the cytoplasm but is not correct as stated.

7. Fig. 4F: The useful statistical comparison for the point mutants would be to the delta-luxR, delta-lsrR strain with the wild-type lsrR plasmid (purple bar).

8. The mice and macrophages used here are C57BL/6J, which is an nramp-negative strain with macrophages deficient in killing Salmonella. Would this have affected the results?

9. Line 327 and following: The AlphaFold3 results are written as if they are certain; the importance of only the Y25 and R43 residues are verified.

Reviewer #2: In my opinion, this work addresses the studied subject with all the necessary experiments, and at this point, I do not recommend any major additional experiments. One aspect I was already considering was covered by the authors in the discussion section. It might be interesting to explore single-cell infection models and examine how co-infection occurs within a single macrophage. Furthermore, since the mutant does not survive in the infected macrophage and SPI-2 is not activated, I would like to see the macrophage polarization after infection, for example, M1/M2 phenotypes.

Reviewer #3: • All along the results:

- the results are often summarised in a single sentence for each figure or supplementary figure subsection. The authors' reasoning is not explained, and it therefore requires considerable effort from the reader to understand why certain experiments were carried out and the results obtained. For example, in the first paragraph, the authors want to demonstrate that AI2 is required for the survival of Salmonella in macrophages (Figure 1E). However, the authors show in Figure 1D that phagocytosis of the luxS, B or K mutants is significantly increased compared to the wt strain, and they give the same value to these two results. It would have been clearer to write that they observed impaired survival of these mutants in macrophages (Figure 1E) and that this reduced survival is not related to an impaired phagocytosis (Figure 1D). By contrast, as expected, the lsrR mutant and the complemented luxS mutant behaved as the wt. It would thus be desirable to expand on the results a little more to explain them better, along with the associated reasoning in most paragraph of results.

- Statistical analyses: except for the in vivo experiments, parametric tests have been performed all along the paper. This assumes that the data are normally distributed. As most of the time n<10, to my opinion, statistical non-parametric tests should have been performed as the Gaussian distribution cannot be insured with this number of data.

- RT-qPCR results: Lines 782-785: no control of the absence of DNA on DNase treated samples is mentioned. Is this an oversight? This step is crucial to ensure a right quantification of RNA. Moreover, only the 16S rRNA gene was used as “housekeeping gene”. The recommendation is to use at least two. Figure 1F and 1G and other graphs showing RT-qPCR results: If the results are expressed as “fold expression of the gene of interest /16S”, why is the ratio obtained always equal to 1 for all genes tested at 2h pi? Moreover, concerning results in Figure S3A,S3B, S4A to S4C, how do you explain that in F acidic medium or in RAW macrophages you obtained the same fold expression/16S as in LB medium (figure 1B and 1C) if AI-2 production is up-regulated at acidic pH. Figure 5A and B: Results obtained on ssrA and ssrB are surprising as these two open reading frames are co-transcribed.

• Macrophage infections: The results are expressed as phagocytosis percentages and proliferation rates. Phagocytosis rates appear to be very low, less than 2%, while proliferation rates are very high (40 to 50) compared to the literature. Was there a transcription error?

• Figure 3: complementation of at least the luxS mutant would improve the results presented in this figure.

• Figure 4E, 4F and figure 6: complementation of the luxS mutant with the phoP gene under a non-native promoter would confirm the results.

• Thijs et al. (https://doi.org/10.1038/cr.2010.104) results should be discussed as they did not found phoP as part of the LsrR regulon

**Part III – Minor Issues: Editorial and Data Presentation Modifications**

Reviewer #1: 10. Some abbreviations are overused, making the manuscript difficult to read. Most notable is STM. As Salmonella Typhimurium is the only organism used here, that can be omitted throughout.

11. Line 147 and elsewhere: The term “luxS complemented STM delta-luxS” is confusing. Adding a hyphen (luxS-complemented) would help.

12. Line 192 and elsewhere: DPD is not defined.

Reviewer #2: The justification and explanation of each part of the results section are too elaborate. Some of it should be placed in either the introduction or the discussion section. This makes the results section too long, and some information is repeated several times throughout the manuscript.

Data presentation of the figures should be better organized, as right now the figures are not aligned properly.

Reviewer #3: • I would recommend to modify the title to introduce Ai-2 and PhoP/Q two-component system

• Introduction The luxS/AI2 system should be presented in the introduction section to facilitate reading by scientists unfamiliar with this system. Figure 3A could be used to illustrate how the system works. Without this information, luxS (line 71), LsrR (lines 88 and 115), the lsr operon (line 117) are not understandable by people not working on the AI2/LuxS system.

• The lsr operon contains only the lsrA, D, C, B, F and G open-reading frames. Not lrsR and lsrK. Please write correctly all over the paper.

• Please replace media by medium all along the manuscript

• First paragraph of the results lines 123 to 180 : This paragraph and figure 1 aim to demonstrate that LuxS/AI2 is required for Salmonella to survive in macrophages. Figures1A,B,C are therefore not required as these results concern bacteria grown in LB medium. I would suggest to remove them and to add figure S1D on figure 1.

• Figure S1E and S1F could not be used to show a decreased ability to proliferate in macrophages. This assertion should be supported by quantification of intracellular Salmonella in several randomly selected fields and in at least three independent experiments.

• Second paragraph of results, lines 181 to 225: In my opinion, these results, which are nevertheless interesting, do not contribute to demonstrating the role of AI-2 in the survival of Salmonella in acidic conditions. Moreover, I do not agree with the conclusion that the luxS mutant cannot acquire AI-2 from a wt strain when intra-vacuolar in macrophages after co-infection of the two strains. Indeed, to conclude this, the authors should provide evidence that wt and mutant strains are simultaneously present in the same vacuole. If not, the mutant could not acquire AI2, because it is not in the same vacuole as the wt and not because it is not able to do so. I would suggest to remove all this paragraph and figure 2A to 2F.

• Third paragraph of the results: Figure 2G could be compared to the results in LB (Figure S1A) to demonstrate the up-regulation of AI-2 production at acidic pH. Perhaps, they can be merged?

• Figure 3C: in which growth medium were the cultures done? Figure 3F, why did the luxS, lsrB and lsrK mutants better survive at pH3 when unadapted at pH 5 compared to adapted at pH5?

• Figure 3H: only two experiments were performed. It is therefore not possible to provide a standard deviation and to perform statistics on these data. At least 3 experiments would be better.

• Line 280: please “compared to the luxS, lsrB and lsrK mutants” after upon exposure to pH3.

• Figure 4 legend: Line 374 please add respectively after promoters

• Lines 461 to 465 : how can you explain the reduced colonization of mice intraperitoneally inoculated with the luxS, lsrB and lsrK mutants

• In vivo, the luxS mutant is as attenuated as the lsrB and lsrK mutants in the intestine. As some members of the microbiota produce AI2, I would have expected that the luxS mutant would be less attenuated than the lsrB and lsrK mutants. Do you have any explanation ?

• Lines 548 and 549: incapable is too strong according to the phenotype you described. Please replace by “less able”

• Lines 573-576: there is a misunderstanding here. SPI-2 effector secretion is triggered by the contact between the T3SS2 and the cytosol of eukaryotic cells (pH neutral) or if bacteria are exposed to pH7 after growth at ph5 (see ref 52), and not because the cytoplasm of bacteria is neutral.

• Line 625: whenever

• AI-2 bioassays: it is not clear how the autoinducer assays were performed on infected macrophages. Was it the cell culture medium, a lysate of macrophages, a lysate of infected macrophages or something else that was used?

• Lines 702-704: not understandable. Proposition: Then 20 μl of Salmonella cell-free culture supernatant was added to 180 μl of this diluted culture of the sensor strain?

• Line 709: control medium

• Line 710 sterile medium

• Lines 756 and 759: please provide the reference of the antibodies and the dilutions used.

• The English of the paper would benefit from some editing.

PLOS authors have the option to publish the peer review history of their article (what does this mean?). If published, this will include your full peer review and any attached files.

Reviewer #1: No

Reviewer #2: No

Reviewer #3: No

**Figure resubmission:**

While revising your submission, we strongly recommend that you use PLOS’s NAAS tool (https://ngplosjournals.pagemajik.ai/artanalysis) to test your figure files. NAAS can convert your figure files to the TIFF file type and meet basic requirements (such as print size, resolution), or provide you with a report on issues that do not meet our requirements and that NAAS cannot fix.  After uploading your figures to PLOS’s NAAS tool - https://ngplosjournals.pagemajik.ai/artanalysis, NAAS will process the files provided and display the results in the "Uploaded Files" section of the page as the processing is complete. If the uploaded figures meet our requirements (or NAAS is able to fix the files to meet our requirements), the figure will be marked as "fixed" above. If NAAS is unable to fix the files, a red "failed" label will appear above. When NAAS has confirmed that the figure files meet our requirements, please download the file via the download option, and include these NAAS processed figure files when submitting your revised manuscript.
---

## [Decision Letter · Decision Letter 1]

16 Mar 2026

PPATHOGENS-D-25-02450R1

LuxS/AI-2 regulates phoP/phoQ by a non-canonical mechanism to enhance acid stress survival in Salmonella Typhimurium

PLOS Pathogens

Dear Dr. Chakravortty,

Thank you for submitting your manuscript to PLOS Pathogens. After careful consideration, we feel that it has merit but does not fully meet PLOS Pathogens's publication criteria as it currently stands. Therefore, we invite you to submit a revised version of the manuscript that addresses the points raised during the review process.

We look forward to receiving your revised manuscript.

Kind regards,

Camila Valenzuela

Guest Editor

PLOS Pathogens

Matthew Wolfgang

Section Editor

PLOS Pathogens

Sumita Bhaduri-McIntosh

Editor-in-Chief

PLOS Pathogens

orcid.org/0000-0003-2946-9497

Michael Malim

Editor-in-Chief

PLOS Pathogens

orcid.org/0000-0002-7699-2064

**Additional Editor Comments :**

Dear Dr. Chakravortty:

Thank you very much for submitting your manuscript "LuxS/AI-2 regulates phoP/phoQ by a non-canonical mechanism to enhance acid stress survival in Salmonella Typhimurium" (PPATHOGENS-D-25-02450R1) for review by PLOS Pathogens. Your revised manuscript was re-evaluated by the same peer reviewers. The reviewers appreciated the improvements and extensive experiments added to the manuscript but identified some aspects of the work that should still be improved. We therefore ask you to revise the manuscript according to the recommendations before we can consider your manuscript for acceptance. Your revisions should address the specific points made by each reviewer.

After reading the reviews and my own appreciation of the the original and the corrected manuscript, although no extra experimental work is required, I would like to note the two main things that still need to be addressed: (1) make sure that all modifications indicated in your last rebuttal are included in the manuscript, all figures are cited and all data is available and (2) that the proper statistical tests are used throughout the manuscript. In particular for this last point, please carefully consider the comments made by Reviewer 3 and the cited literature on this regards. After these items are addressed, I will be sending the manuscript for a final and second round of review. Please pay particular attention to the major and minor changes reviewer 3 suggested and give them due consideration. In addition, and to facilitate the next round of reviewing, please indicate in the rebuttal the lines that now contain the corresponding changes in the updated manuscript.

When you are ready to resubmit, please be prepared to provide the following:(1) A letter containing a detailed list of your responses to the review comments and a description of the changes you have made in the manuscript. (2) Two versions of the manuscript: one with either highlights or tracked changes denoting where the text has been changed; the other a clean version (uploaded as the manuscript file).

We hope to receive your revised manuscript within 60 days or less. If you anticipate any delay in its return, we ask that you let us know the expected resubmission date by replying to this email. If you have any questions or concerns while you make these revisions, please let us know.

Sincerely,

Camila Valenzuela, PhD

Guest Editor

PLOS Pathogens

Matthew C. Wolfgang, Ph.D.

Section Editor

PLOS Pathogens

**Journal Requirements:**

1) Please upload a copy of Figures 2, and 7 which you refer to in your text. Or, if the figures are no longer to be included as part of the submission please remove all reference to them within the text.

2) We have noticed that you have referred to supplementary tables and "Supplementary method" in the list of Supporting Information legends in your manuscript. However, there are no corresponding files uploaded to the submission. Please upload them as separate files with the item type 'Supporting Information'.

3) Please ensure that the Data Availability Statements provided in the submission form and the manuscript are the same.

4) Please amend your detailed Financial Disclosure statement. This is published with the article. It must therefore be completed in full sentences and contain the exact wording you wish to be published.

5) Please ensure that the funders and grant numbers match between the Financial Disclosure field and the Funding Information tab in your submission form. Note that the funders must be provided in the same order in both places as well.

**Reviewers' Comments:**

Reviewer's Responses to Questions

**Part I - Summary**

Reviewer #1: The authors have addressed the comments of my previous review.

Reviewer #2: The authors addressed all my concerns, and I claim that the current version of the publication is almost ready for acceptance.

Reviewer #3: I would like to thank the authors for their point-by-point answers to the Reviewer comments. Answers are very complete. I also really appreciate all the experiments that have been performed and added to the manuscript. My questions and concerns are listed below.

**Part II – Major Issues: Key Experiments Required for Acceptance**

Reviewer #1: None

Reviewer #2: The are no additional experiment needed.

Reviewer #3: Major issues:

After careful reading of the manuscript, I am not sure that all the modifications mentioned by the authors in their answers have been included in the paper. I wonder if the version provided is the correct one (I read the yellow highlighted version) and I apologize if I missed several modifications. A few examples concern the major issues of (1) Reviewer 1 point 2 (I did not find changes announced by the authors in the discussion concerning this point), (2) Reviewer 2 issues on figure 2G and line 234 (comparison not found in the results section) or on macrophages polarisation (did not find anything on this point in the discussion section) (3) Reviewer 3 “Control of DNA absence in RNA samples” (where in the method section?) (4) Reviewer 3 normalization relative to 2h (legends and methods). Please make sure that all the modifications asked by the reviewers have been introduced.

Statistical analyses: I am sorry but I do not really agree with the author’s answer to my concern about their statistical analyses. I agree with the authors that nonparametric tests are generally considered less powerful than parametric tests, but this does not justify the use of parametric tests if the assumptions for these latter tests are not met. The authors mention that they used parametric tests because their data follow a normal distribution (“symmetric”). How did the authors determine the normality of their data, given that statistical tests of normality are only useful for large samples (Van Belle G, Fisher LD, Heagerty PJ, Lumley T. Biostatistics: A Methodology for the Health Sciences. Hoboken, NJ: Wiley, 2004.)? Did they use normal probability plots? This could be a way to test normality for low sample size (Morgan CJ https://doi.org/10.1152/ajplung.00238.2017). Otherwise, permutation tests can be used, which appear to be as powerful as parametric tests and are suitable without the validity of the results relying on theoretical distributions.

I am not surprised that other papers in your research field used similar statistical tests as those you did. Most of us have used parametric tests incorrectly or without testing if the assumptions for these tests were met at some point in our careers. I believe that this should change for all of us. Please read the paper of (Olsen CH. 2014. Statistics in Infection and Immunity Revisited. Infect Immun 82:https://doi.org/10.1128/iai.00811-13), which provides a good overview of good and bad practices in statistics applied to our field of research.

**Part III – Minor Issues: Editorial and Data Presentation Modifications**

Reviewer #1: None

Reviewer #2: The authors address all of my concerns. However, I have a question about the results and the statistical test used. There are many figures and experiments in which the authors compare 6 or 7 groups, with only three results per group, and they use ANOVA to do so. As far as I am concerned, to use ANOVA properly, you need at least N+1 biological replicates; for example, when comparing 7 groups, you should have at least 8 samples per group. Can the authors address my concern??

Reviewer #3: - If I am not mistaken, Supplementary tables are not provided into the pdf file provided to the Reviewers. It is thus not possible to verify that the new strains and oligos used to answer the reviewer comments have been added.

- Figure 7 is not cited in the text. It seems to correspond to a Graphical abstract but I do not see any graphical abstract in papers published in PLOS Pathogens. Is it new ? Is a legend necessary?

- Data availability statement: the authors have revised this section and state that all relevant data are within the manuscript and it Supporting Information files”. I am not sure this is the case. In most legends, the number of experiments and replicates is indicated (e.g., N=3 n=3) but the number of corresponding values is not provided in the figures (i.e. 9 values). Does this mean that figures were build using representative experiments, that some data were excluded or that all values are not presented and should therefore be added? Please look at all the legends and the corresponding figure or supplementary figure and provide an explanation for each of them or provide the missing data.

- Is S1B Fig cited somewhere? I did not find it in the text.

- Fig 1A “Regulate multiple QS genes” instead of Regulate multiples genes-QS genes” and “phosphorylated AI-2 binds LsrR and removes it from lsr promoter” instead of “phosphorylated AI-2 binds LsrR and remove it from lsr promoter”

- Concerning my comment on the fold expression to 1 at 2h post-infection. Thank you for the explanations that are clear. In this case, I would suggest to change the title of the y axis of the graphs of the corresponding figures. For example, “relative expression of “gene X”/2h pi”

- I would like to thank the authors for including the paper of Thijs in the discussion part. According to EMSA results, it seems to me that the authors observe LsrR binding to both the single and the double-stranded DNA of the phoPQ promoter. Am I wrong? If not, I am not sure that the single strand interaction observed by the authors could explain the difference obtained in the two studies.

- Line 117: please add that DPD is secreted by Salmonella.

- Lines 171-174 : I agree that in S1G figure, the data suggest that bacteria reside in a vacuole as LAMP1 colocalize with the strains. I am not convinced for the S1H figure, especially for the lsrR mutant. Numerous bacteria do not colocalize with LAMP1. This picture looks like cytosolic hyper-replicative bacteria that have already been described in RAW 264.7 cells by Roder and Hensel 2020(DOI: 10.1111/cmi.13155). As only a few cells are shown, we cannot exclude that cytosolic hyper-replication could also be observed for the other strains as this phenomenon is rare (about 10% of the infected cells).

- Line 177 please add compared to 2h pi

- Figure 2 legend: for H, I and J please provide N and n

- In my opinion, Figure 2G does not allow to conclude that acidification is a physiological signal that upregulates AI-2 production by Samonella (line 265). A comparison with data obtained after growth at neutral pH in the same medium is needed. Please clarify the use of the term “up-regulation” in line 265.

Moreover, the experiments performed with Vibrio allow the amount of AI-2 in the supernatant of Salmonella culture to be evaluated. I agree that the amount of AI2 present in the supernatant is maximal during the mid- and late-log phases but the amount of AI2 produced by Salmonella is probably not maximal at these two phases, as the number of bacteria secreting AI2 in the supernatant is higher during these growth phases than in the lag-phase and in the early logarithmic phase. Therefore, I am not sure that the statement “Moreover, we observed that STM stably produces AI2 even during its growth in F-medium, which is maximal during the mid to late-log phase” is true. In my opinion, it is important to distinguish between the amount of AI2 present in the supernatant and the amount of AI2 produced by Salmonella. Expressing the results in RLU/OD, as requested by another reviewer, answers the question of how much AI2 is produced by Salmonella and is a complementary information to the amount of AI2 present in the culture supernatant.

- Figure S4 and Lines 296 to 304: The different parts of this figure are not correctly referenced in the text. Figures 4A to 4C do not compare the WT and the mutants, Figure 4D is not related to F-medium, Figure 4G is not a comparison of phoP and phoQ expression in the WT and mutants upon infection of macrophages. Please correctly reassign the figures to the corresponding text.

- Line 324: measured instead of validated

- Line 325: please add “and STM WT “

- Line 326: please add “and observed diminished…”

- Fig 3 B legend: please precise the medium used

- Line 388: According to quantifications provided in FigS6, Y25A alone and R43A alone do not seem to reduce the binding of LsrR compared to WT LsrR in most cases. What data do the authors base the assertion that Y25A and R43A showed reduced binding on?

- Lines 392-394 and Lines 395-397: duplicate entry?

- Figure 4 legend: C and D please specify if EMSA were performed with double or single strand DNA.

- Figure S10 and S11: what does gm (y axis) means? And what is shown the Mice weight reduction (S10 legend) or mice weight (S11 legend and S10E-F and S11F figures)?

PLOS authors have the option to publish the peer review history of their article (what does this mean?). If published, this will include your full peer review and any attached files.

Reviewer #1: No

Reviewer #2: No

Reviewer #3: No

**Figure resubmission:**
---

## [Decision Letter · Decision Letter 2]

28 Apr 2026

PPATHOGENS-D-25-02450R2

LuxS/AI-2 regulates phoP/phoQ by a non-canonical mechanism to enhance acid stress survival in Salmonella Typhimurium

PLOS Pathogens

Dear Dr. Chakravortty,

Thank you for submitting your manuscript to PLOS Pathogens. After careful consideration, we feel that it has merit but does not fully meet PLOS Pathogens's publication criteria as it currently stands. Therefore, we invite you to submit a revised version of the manuscript that addresses the points raised during the review process.

We look forward to receiving your revised manuscript.

Kind regards,

Camila Valenzuela

Guest Editor

PLOS Pathogens

Matthew Wolfgang

Section Editor

PLOS Pathogens

Sumita Bhaduri-McIntosh

Editor-in-Chief

PLOS Pathogens

orcid.org/0000-0003-2946-9497

Michael Malim

Editor-in-Chief

PLOS Pathogens

orcid.org/0000-0002-7699-2064

**Additional Editor Comments:**

Dear Dr. Chakravortty:

Thank you very much for your re-revised version of the manuscript "LuxS/AI-2 regulates phoP/phoQ by a non-canonical mechanism to enhance acid stress survival in Salmonella Typhimurium" (PPATHOGENS-D-25-02450R2). Your revised manuscript was re-evaluated by two of the original peer reviewers, who agreed that the modifications made address almost all aspects discussed during the previous rounds of revision. After carefully reading the minor comments from Reviewer 3, I have come to the conclusion that these data visualization changes should be incorporated before we can formally accept this manuscript. Therefore, I am recommending a Minor Revision, after which I will personally evaluate how the changes were implemented, without sending it to the original reviewers, and provisionally accept the new version.

When you are ready to resubmit, please be prepared to provide the following:

(1) A rebuttal letter containing with a short description of the changes you have made in the manuscript.

(2) Two versions of the manuscript: one with either highlights or tracked changes denoting where the text and figures have been changed; the other a clean version (uploaded as the manuscript file).

I understand that this might sound like a delay, but I am sure these small changes can be addressed in one or two weeks, after which I will be happy to promptly accept your manuscript for publication in PLOS Pathogens.

Thank you again for your interest in and support for PLOS Pathogens and open access publishing.

Best wishes, Camila Valenzuela, PhD

Guest Editor

PLOS Pathogens

**Journal Requirements:**

1) We have noticed that you have uploaded Supporting Information files, but you have not included a complete list of legends. Please add a full list of legends for your Supporting Information file (Supporting Information.docx) after the references list.

2) Please ensure that the funders and grant numbers match between the Financial Disclosure field and the Funding Information tab in your submission form. Note that the funders must be provided in the same order in both places as well.

**Reviewers' Comments:**

Reviewer's Responses to Questions

**Part I - Summary**

Reviewer #2: In my opinion the paper is ready for acceptance.

Reviewer #3: I would like to thank the authors for taking into account all the Reviewer comments and for their detailed answers in this R2 version. I have now only a few minor concerns

**Part II – Major Issues: Key Experiments Required for Acceptance**

Reviewer #2: none.

Reviewer #3: None

**Part III – Minor Issues: Editorial and Data Presentation Modifications**

Reviewer #2: none

Reviewer #3: • The availability of the data is not still very clear to me, even though many details have been added. I would like to give a few examples, but this issue recurs throughout the article:

- In Figure S5 ABC, there are only three data points per histogram, whereas in the legend, the authors wrote, “All data are represented as… (N=3, n=3).”

- In Figure 1E, numerous points are shown on each histogram, but in the legend, the authors wrote “the data come from an experiment representative,” which implies that only 3 points should be present,

- In Figures 1F and 1H, only 3 points are present for each histogram, but upon reading the legend, it seems to me that all the data should have been provided.

- In Figure 2, I do not see any histogram with more than 4 data points, but in the corresponding legend, the authors did not always write “The data are from a representative experiment of 3 independent experiments, N=3, n=4.” Instead, at the end of the legend for Figure 2, they wrote, “All data are presented as mean ± standard deviation from three independent experiments.”

• Regarding Figure S1H and the colocalization (or not of Salmonella) with LAMP-1, I do not share the authors’ view. Rôder and Hensel 2020 did not work exclusively with epithelial HeLa cells in their paper as stated by the authors. They also tested RAW 264.7 macrophages and wrote: “Time-lapse microscopy of RAW264.7 cells infected with STM harbouring the dual colour reporter revealed intracellular proliferation of STM WT (Movie 3) and occasionally the appearance of host cells with sfGFP-expressing cytosolic bacteria … (Movie 4)”. These cytosolic Salmonella are observable using confocal microscopy thus under the conditions used by Singh et al.. We confirmed the results of Röder and Hensel on RAW 264.7 cells in our lab using both the reporter plasmid used by Röder and Hensel and LAMP-1 and Salmonella colocalization experiments, as Singh et al did.

The S1H picture shows a lack of LAMP-1 colocalization with numerous Salmonella for the lsrR mutant and a typical picture of cytosolic hyper-replication we observe when performing this type of experiment on RAW 264.7 experiments. I therefore agree with the authors that “Salmonella adopt a predominantly vacuolar life style within macrophages”, but Salmonella also adopts a cytosolic lifestyle in macrophages. The lack of LAMP-1 colocalization with Salmonella in figure S1H for the lsrR mutant therefore does not allow to conclude that Salmonella is vacuolar for this mutant (Line 173-174). And, as I mentioned in my review of version R1, “As only a few cells are shown in FigS1H for all the strains tested, we cannot exclude that cytosolic hyper-replication could also be observed for other strains than the lsrR mutant as cytosolic Salmonella is rare phenomenon (about 10% of the infected cells in our hands)”. In conclusion, in my opinion, the authors cannot claim that “all the strains reside in SCV” (Lines 173-174) based on the confocal microscopy images presented in figure S1H.

• Line 58 : please do not italicize Typhimurium

• Line 136: please remove one of the two “the and add the notion that LsrR binds to both single and double strand phoP/phoQ promoter DNA

• Line 269 I suppose that the authors wanted to write “Our AI-2 bioassay in acidic medium shows AI-2 production by Salmonella

• Line 275-279: please rewrite which is long and not clear.

• Line 301303: Thank you for having added FigS4A-D. Please please add at least a short interpretation of the results shown in these figures

• FigS2C: I would suggest to add DPD after the μM concentration on the x axis

• FigS2I x axis: I would suggest to write BF-8 instead of BF

• Table S1: please provide bibliographic reference instead of “Laboratory stock” as sources in Table S1 so that scientists reading your paper can access the history of the plasmids and strains you used

• Fig S10 and 11: if the weight is expressed in grams, then I would suggest to use the international nomenclature i.e. g instead of gm

PLOS authors have the option to publish the peer review history of their article (what does this mean?). If published, this will include your full peer review and any attached files.

Reviewer #2: No

Reviewer #3: No

**Figure resubmission:**
---

## [Editor Report · Decision Letter 3]

6 May 2026

Dear Prof. Chakravortty,

We are pleased to inform you that your manuscript 'LuxS/AI-2 regulates phoP/phoQ by a non-canonical mechanism to enhance acid stress survival in Salmonella Typhimurium' has been provisionally accepted for publication in PLOS Pathogens.

Best regards,

Camila Valenzuela

Guest Editor

PLOS Pathogens

David Skurnik

Section Editor

PLOS Pathogens

Sumita Bhaduri-McIntosh

Editor-in-Chief

PLOS Pathogens

orcid.org/0000-0003-2946-9497

Michael Malim

Editor-in-Chief

PLOS Pathogens

orcid.org/0000-0002-7699-2064

Dear Dr. Chakravortty,

Thank you for your revised version of the manuscript "LuxS/AI-2 regulates phoP/phoQ by a non-canonical mechanism to enhance acid stress survival in Salmonella Typhimurium" (PPATHOGENS-D-25-02450R3). Your revised manuscript was evaluated by me and after assessing that the modifications made address all aspects discussed during the revision, we are pleased to inform you that your manuscript has been provisionally accepted for publication in PLOS Pathogens.

Thank you again for your interest in and support for PLOS Pathogens and open access publishing.

Best wishes,

Camila Valenzuela, PhD

Guest Editor

PLOS Pathogens

Matthew C. Wolfgang, Ph.D

Section Editor

PLOS Pathogens
---

## [Editor Report · Acceptance letter]

Dear Prof. Chakravortty,

We are delighted to inform you that your manuscript, "LuxS/AI-2 regulates phoP/phoQ by a non-canonical mechanism to enhance acid stress survival in Salmonella Typhimurium," has been formally accepted for publication in PLOS Pathogens.

Best regards,

Sumita Bhaduri-McIntosh

Editor-in-Chief

PLOS Pathogens

orcid.org/0000-0003-2946-9497

Michael Malim

Editor-in-Chief

PLOS Pathogens

orcid.org/0000-0002-7699-2064